# Jump-Start Reinforcement Learning

## Abstract

Reinforcement learning (RL) provides a theoretical framework for continuously improving an agent's behavior via trial and error. However, efficiently learning policies from scratch can be very difficult, particularly for tasks that present exploration challenges. In such settings, it might be desirable to initialize RL with an existing policy, offline data, or demonstrations. However, naively performing such initialization in RL often works poorly, especially for value-based methods. In this paper, we present a meta algorithm that can use offline data, demonstrations, or a pre-existing policy to initialize an RL policy, and is compatible with any RL approach. In particular, we propose Jump-Start Reinforcement Learning (JSRL), an algorithm that employs two policies to solve tasks: a guide-policy, and an exploration-policy. By using the guide-policy to form a curriculum of starting states for the exploration-policy, we are able to efficiently improve performance on a set of simulated robotic tasks. We show via experiments that it is able to significantly outperform existing imitation and reinforcement learning algorithms, particularly in the small-data regime. In addition, we provide an upper bound on the sample complexity of JSRL and show that with the help of a guide-policy, one can improve the sample complexity for non-optimism exploration methods from exponential in horizon to polynomial.

## 1 Introduction

A promising aspect of reinforcement learning (RL) is the ability of a policy to iteratively improve via trial and error. Often, however, the most difficult part of this process is the very beginning, where a policy that is learning without any prior data needs to randomly encounter rewards to further improve. A common way to side-step this exploration issue is to aid the policy with prior knowledge. One source of prior knowledge might come in the form of a prior policy, which can provide some initial guidance in collecting data with non-zero rewards, but which is not by itself fully optimal. Such policies could be obtained from demonstration data (e.g., via behavioral cloning), from sub-optimal prior data (e.g., via offline RL), or even simply via manual engineering. In the case where this prior policy is itself parameterized as a function approximator, it could serve to simply initialize a policy gradient method. However, sample-efficient algorithms based on value functions are notoriously difficult to bootstrap in this way. As observed in prior work (Peng et al., 2019; Nair et al., 2020; Kostrikov et al., 2021; Lu et al., 2021), value functions require both good and bad data to initialize successfully, and the mere availability of a starting policy does not by itself readily provide an initial value function of comparable performance. This leads to the question we pose in this work: how can we bootstrap a value-based RL algorithm with a prior policy that attains reasonable but sub-optimal performance?

The main insight that we leverage to address this problem is that we can bootstrap any RL algorithm by gradually "rolling in" with the prior policy, which we refer to as the guide-policy. In particular, the guide-policy provides a curriculum of starting states for the RL exploration-policy, which significantly simplifies the exploration problem and allows for fast learning. As the exploration-policy improves, the effect of the guide-policy is diminished, leading to an RL-only policy that is capable of further autonomous improvement. Our approach is generic, as it can be applied to any RL method that explores its environment for policy improvement, though we focus on value-based methods in this work. The only requirements of our method are that the guide-policy can select actions based on observations of the environment, and its performance is reasonable (i.e., better than a random

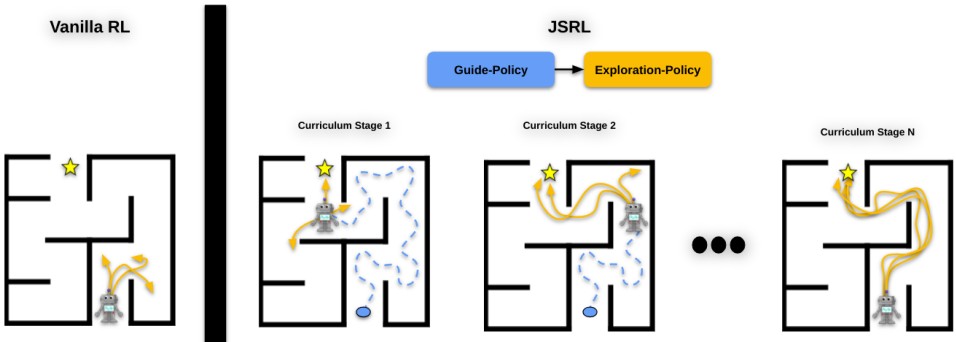

Figure 1: We study how to efficiently bootstrap value-based RL algorithms given access to a prior policy. In vanilla RL (left), the agent explores randomly from the initial state until it encounters a reward (gold star). JSRL (right), leverages a guide-policy (dashed blue line) that takes the agent closer to the reward. After the guide-policy finishes, the exploration-policy (solid orange line) continues acting in the environment. As the exploration-policy improves, the influence of the guide-policy diminishes, resulting in a learning curriculum for bootstrapping RL.

policy). Since the guide-policy significantly speeds up the early phases of RL, we call this approach Jump-Start Reinforcement Learning (JSRL). We provide an overview diagram of JSRL in Fig. 1.

JSRL can utilize any form of prior policy to accelerate RL. It is also compatible with RL algorithms that involve rolling out a policy to explore an environment. Thus, JSRL can easily be combined with existing offline and/or online RL methods. In addition, we provide a theoretical justification of JSRL by deriving an upper bound on its sample complexity compared to RL alternatives. Finally, we demonstrate that JSRL outperforms previously proposed imitation and reinforcement learning approaches on a set of benchmark tasks as well as more challenging vision-based robotic problems.

## 2 RELATED WORK

**Imitation learning combined with reinforcement learning (IL+RL)**. Several previous works on leveraging a prior policy to initialize RL focus on doing so by combining imitation learning and RL. Some methods treat RL as a sequence modelling problem and train an autoregressive model using offline data Zheng et al. (2022); Janner et al. (2021); Chen et al. (2021). One well-studied class of approaches initializes policy search methods with policies trained via behavioral cloning Schaal et al. (1997); Kober et al. (2010); Rajeswaran et al. (2017). This is an effective strategy for initializing policy search methods, but is generally ineffective with actor-critic or value-based methods, where the critic also needs to be initialized (Nair et al., 2020), as we also illustrate in Section 3. Methods have been proposed to include prior data in the replay buffer for a value-based approach (Nair et al., 2018; Vecerik et al., 2018), but this requires prior *data* rather than just a prior *policy*. More recent approaches improve this strategy by using offline RL Kumar et al. (2020); Nair et al. (2020); Lu et al. (2021) to pre-train on prior data, then finetune. We compare to such methods, showing that our approach not only makes weaker assumptions (requiring only a policy rather than a dataset), but also performs comparably or better.

**Curriculum learning and exact state resets for RL**. Many prior works have investigated efficient exploration strategies in RL that are based on starting exploration from specific states. Commonly, these works assume the ability to reset to arbitrary states in simulation (Salimans & Chen, 2018). Some methods uniformly sample states from demonstrations as start states (Hosu & Rebedea, 2016; Peng et al., 2018; Nair et al., 2018), while others generate curriculas of start states. The latter includes methods that start at the goal state and iteratively expand the start state distribution, assuming reversible dynamics (Florensa et al., 2017; McAleer et al., 2019) or access to an approximate dynamics model (Ivanovic et al., 2019). Other approaches generate the curriculum from demonstration states (Resnick et al., 2018) or from online exploration (Ecoffet et al., 2021). In contrast, our method does not control the exact starting state distribution, but instead utilizes the implicit distribution

---

[0]A project webpage is available at https://jumpstartrl.github.io

naturally arising from rolling out the guide-policy. This broadens the distribution of start states compared to exact resets along a narrow set of demonstrations, making the learning process more robust. In addition, our approach could be extended to the real world, where resetting to a state in the environment is impossible.

**Provably efficient exploration techniques.** Online exploration in RL has been well studied in theory (Osband & Van Roy, 2014; Jin et al., 2018; Zhang et al., 2020b; Xie et al., 2021; Zanette et al., 2020; Jin et al., 2020). The proposed methods either rely on the estimation of confidence intervals (e.g. UCB, Thompson sampling), which is hard to approximate and implement when combined with neural networks, or suffer from exponential sample complexity in the worst-case. In this paper, we leverage a pre-trained guide-policy to design an algorithm that is more sample-efficient than these approaches while being easy to implement in practice.

**"Rolling in" policies.** Using a pre-existing policy (or policies) to initialize RL and improve exploration has been studied in past literature. Some works use an ensemble of roll-in policies or value functions to refine exploration Jiang et al. (2017); Agarwal et al. (2020). With a policy that models the environment's dynamics, it is possible to look ahead to guide the training policy towards useful actions (Lin, 1992). Similar to our work, an approach from Smart & Pack Kaelbling (2002) rolls out a fixed controller to provide bootstrap data for a policy's value function. However, this method does not mix the prior policy and the learned policy, but only uses the prior policy for data collection. We use a multi-stage curriculum to gradually reduce the contribution of the prior policy during training, which allows for on-policy experience for the learned policy. Our method is also conceptually related to DAgger (Ross & Bagnell, 2010), which also bridges distributional shift by rolling in with one policy and then obtaining labels from a human expert, but DAgger is intended for imitation learning and rolls in the learned policy, while our method addresses RL and rolls in with a sub-optimal guide-policy.

## 3 PRELIMINARIES

We define a Markov decision process $\mathcal{M} = (\mathcal{S}, \mathcal{A}, P, R, p_0, \gamma, H)$, where $\mathcal{S}$ and $\mathcal{A}$ are state and action spaces, $P : \mathcal{S} \times \mathcal{A} \times \mathcal{S} \to \mathbb{R}_+$ is a state-transition probability function, $R : \mathcal{S} \times \mathcal{A} \to \mathbb{R}$ is a reward function, $p_0 : \mathcal{S} \to \mathbb{R}_+$ is an initial state distribution, $\gamma$ is a discount factor, and $H$ is the task horizon. Our goal is to effectively utilize a prior policy of any form in value-based reinforcement learning (RL). The goal of RL is to find a policy $\pi(a|s)$ that maximizes the expected discounted reward over trajectories, $\tau$, induced by the policy: $\mathbb{E}_\pi[R(\tau)]$ where $s_0 \sim p_0, s_{t+1} \sim P(\cdot|s_t, a_t)$ and $a_t \sim \pi(\cdot|s_t)$. To solve this maximization problem, value-based RL methods take advantage of state or state-action value functions (Q-function) $Q^\pi(s, a)$, which can be learned using approximate dynamic programming approaches. The Q-function, $Q^\pi(s, a)$, represents the discounted returns when starting from state $s$ and action $a$, followed by the actions produced by the policy $\pi$.

In order to leverage prior data in value-based RL and continue fine-tuning, researchers commonly use various offline RL methods (Kostrikov et al., 2021; Kumar et al., 2020; Nair et al., 2020; Lu et al., 2021) that often rely on pre-trained, regularized Q-functions that can be further improved using online data. In the case where a pre-trained Q-function is not available and we only have access to a prior policy, value-based RL methods struggle to effectively incorporate that information as depicted in Fig. 2. In this experiment, we train an actor-critic method up to step 0, then we start from a fresh Q-function and

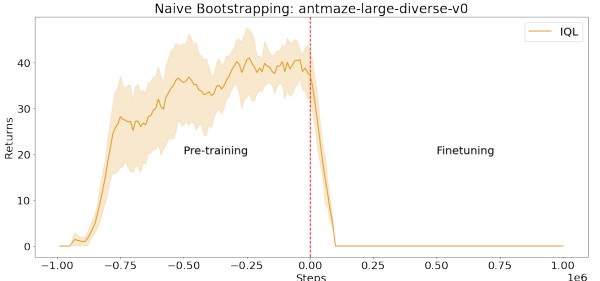

Figure 2: **Naïve policy initialization.** We pre-train a policy to medium performance (depicted by negative steps), then use this policy to initialize actor-critic fine-tuning (starting from step 0), while initializing the critic randomly. Actor performance decays, as the untrained critic provides a poor learning signal, causing the good initial policy to be forgotten. In Figures 7 and 8, we repeat this experiment but allow the randomly initialized critic to "warm up" before fine-tuning.

continue with the pre-trained actor, simulating the case where we only have access to a prior policy. This is the setting that we are concerned with in this work.

# 4 JUMP-START REINFORCEMENT LEARNING

In this section, we describe our method, Jump-Start Reinforcement Learning (JSRL), that we use to initialize value-based RL algorithms with a prior policy of any form. We first describe the intuition behind our method then lay out a detailed algorithm along with theoretical analysis.

## 4.1 ROLLING IN WITH TWO POLICIES

We assume access to a fixed prior policy that we refer to as the "guide-policy", $\pi^g(a|s)$, which we leverage to initialize an RL algorithm. It is important to note that we do not assume any particular form of $\pi^g$; it could be learned with imitation learning, RL, or it could be manually scripted. We will refer to the RL policy that is being learned via trial and error as the "exploration-policy" $\pi^e(a|s)$, since, as it is commonly done in RL literature, this is the policy that is used for exploration as well as online improvement. The only requirement for $\pi^e$ is that it is an RL policy that can adapt with online experience. Our approach and the set of assumptions is generic in that it can handle any downstream RL method that rolls out a policy for exploring an environment, though we focus on the case where $\pi^e$ is learned via a value-based RL algorithm.

The main idea behind our method is to leverage the two policies, $\pi^g$ and $\pi^e$, executed sequentially to learn tasks more efficiently. During the initial phases of training, $\pi^g$ is significantly better than the untrained policy $\pi^e$, so we would like to collect data using $\pi^g$. However, this data is *out of distribution* for $\pi^e$, since exploring with $\pi^e$ will visit different states. Therefore, we would like to gradually transition data collection away from $\pi^g$ and toward $\pi^e$. Intuitively, we would like to use $\pi^g$ to get the agent into "good" states, and then let $\pi^e$ take over and explore from those states. As it gets better and better, $\pi^e$ should take over earlier and earlier, until all data is being collected by $\pi^e$ and there is no more distributional shift. We can employ different switching strategies to switch from $\pi^g$ to $\pi^e$, but the most direct curriculum simply switches from $\pi^g$ to $\pi^e$ at some time step $h$, where $h$ is initialized to the full task horizon and gradually decreases over the course of training. This naturally provides a curriculum for $\pi^e$. At each curriculum stage, $\pi^e$ needs to master a small part of the state-space that is required to reach the states covered by the previous curriculum stage.

## 4.2 ALGORITHM

We provide a detailed description of JSRL in Algorithm 1. Given an RL task with horizon $H$, we first choose a sequence of initial guide-steps to which we roll out our guide-policy, $\{H_1, H_2, \cdots, H_n\}$, where $H_i \in \{1, 2, \cdots, H\}$ denotes the number of steps that the guide-policy at the $i$th iteration acts for. Let $h$ denote the iterator over such a sequence of initial guide-steps. At the beginning of each training episode, we roll out $\pi^g$ for $h$ steps, then $\pi^e$ continues acting in the environment for the additional $H - h$ steps until the task horizon $H$ is reached. We can write the combination of the two policies as the combined policy, $\pi$, where $\pi_{1:h} = \pi^g$ and $\pi_{h+1:H} = \pi^e$. After we roll out $\pi$ to collect online data, we use the new data to update our exploration-policy $\pi^e$ and combined policy $\pi$ by calling a standard training procedure TRAINPOLICY. The TRAINPOLICY updates both the $Q$ function and the corresponding evaluation policy. For example, the training procedure may be updating the exploration-policy via a Deep Q-Network (Mnih et al., 2013) with $\epsilon$-greedy as the exploration technique (i.e. $\pi^e(a|s) = 1 - \epsilon$ if $a = \arg\max_a Q(s, a)$ and $\epsilon/|\mathcal{A}|$ otherwise). The new combined policy is then evaluated over the course of training using a standard evaluation procedure EVALUATEPOLICY($\pi$). Once the performance of the combined policy $\pi$ reaches a threshold, $\beta$, we continue the for loop with the next guide step $h$.

While any guide-step sequence could be used with JSRL, we focus on two specific strategies for determining guide-step sequences: curriculum and random-switching. With the curriculum strategy, we start with a large guide-step (ie. $H_1 = H$) and use policy evaluations of the combined policy $\pi$ to progressively decrease $H_n$ as $\pi^e$ improves. Intuitively, this means that we train our policy in a backward manner by first rolling out $\pi^g$ to the last guide-step and then exploring with $\pi^e$, and then rolling out $\pi^g$ to the second to last guide-step and exploring with $\pi^e$, and so on. With the random-switching strategy, we sample each $h$ uniformly and independently from the set $\{H_1, H_2, \cdots, H_n\}$.

In the rest of the paper, we refer to the curriculum variant as JSRL, and the random switching variant as JSRL-Random.

---
**Algorithm 1** Jump-Start Reinforcement Learning
---
1: **Input:** guide-policy $\pi^g$, performance threshold $\beta$, task horizon $H$, a sequence of initial guide-steps $H_1, H_2, \cdots, H_n$, where $H_i \in \{1, 2, \cdots, H\}$ for all $i \leq n$.
2: Initialize exploration-policy from scratch or with the guide-policy $\pi^e \leftarrow \pi^g$. Initialize $Q$-function $\hat{Q}$ and dataset $\mathcal{D} \leftarrow \varnothing$.
3: **for** current guide step $h = H_1, H_2, \cdots, H_n$ **do**
4:      Set the non-stationary policy $\pi_{1:h} = \pi^g$, $\pi_{h+1:H} = \pi^e$
5:      Roll out the policy $\pi$ to get trajectory $\{(s_1, a_1, r_1), \cdots, (s_H, a_H, r_H)\}$; Append the trajectory to the dataset $\mathcal{D}$.
6:      $\pi^e, \hat{Q} \leftarrow \text{TRAINPOLICY}(\pi^e, \hat{Q}, \mathcal{D})$
7:      **if** $\text{EVALUATEPOLICY}(\pi) \geq \beta$ **then**
8:          Continue
9:      **end if**
10: **end for**

---

### 4.3 THEORETICAL ANALYSIS

In this section, we provide theoretical analysis of JSRL, showing that the roll-in data collection strategy that we propose provably attains polynomial sample complexity. The sample complexity refers to the number of samples required by the algorithm to learn a policy with small suboptimality, where we define the suboptimality for a policy $\pi$ as $\mathbb{E}_{s \sim p_0}[V^\star(s) - V^\pi(s)]$. In particular, we aim to answer two questions: *Why is JSRL better than other exploration algorithms which start exploration from scratch? Under which conditions does the guide-policy provably improve exploration?* To answer these questions, we study upper and lower bounds for the sample complexity of exploration algorithms. We first provide a lower bound showing that simple non-optimism-based exploration algorithms like $\epsilon$-greedy suffer from a sample complexity that is exponential in the horizon. Then, we show that with the help of a guide-policy with good coverage of important states, the JSRL algorithm with $\epsilon$-greedy as the exploration strategy can achieve polynomial sample complexity.

We focus on comparing JSRL with standard non-optimism-based exploration methods, e.g. $\epsilon$-greedy (Langford & Zhang, 2007) and FALCON+ (Simchi-Levi & Xu, 2020). Although the optimism-based RL algorithms like UCB (Jin et al., 2018) and Thompson sampling (Ouyang et al., 2017) turn out to be efficient strategies for exploration from scratch, they all require uncertainty quantification, which can be hard for vision-based RL tasks with neural network parameterization. Note that the cross entropy method used in the vision-based RL framework Qt-Opt (Kalashnikov et al., 2018) is also a non-optimism-based method. In particular, it can be viewed as a variant of $\epsilon$-greedy algorithm in continuous action space, with the Gaussian distribution as the exploration distribution.

We first show that without the help of a guide-policy, the non-optimism-based method usually suffers from a sample complexity that is exponential in horizon for episodic MDP. We adapt the combination lock example in Koenig & Simmons (1993) to show the hardness of exploration from scratch for non-optimism-based methods.

**Theorem 4.1** (Koenig & Simmons (1993)). *For 0-initialized $\epsilon$-greedy, there exists an MDP instance such that one has to suffer from a sample complexity that is exponential in total horizon $H$ in order to find a policy that has suboptimality smaller than $0.5$.*

We include the construction of combination lock MDP and the proof in Appendix A.4.2 for completeness. This lower bound also applies to any other non-optimism-based exploration algorithm which explores uniformly when the estimated $Q$ for all actions are $0$. As a concrete example, this also shows that iteratively running FALCON+ Simchi-Levi & Xu (2020) suffers from exponential sample complexity. With the above lower bound, we are ready to show the upper bound for JSRL under certain assumptions on the guide-policy. In particular, we assume that the guide-policy $\pi^g$ is able to cover good states that are visited by the optimal policy under some feature representation:

**Assumption 4.2** (Quality of guide-policy $\pi^g$). Let $d^\pi(s)$ be the marginalized state occupancy distribution when we follow policy $\pi$. Assume that the state is parametrized by some feature mapping $\phi : \mathcal{S} \mapsto \mathbb{R}^d$ such that for any policy $\pi$, $Q^\pi(s, a)$ and $\pi(s)$ depend on $s$ only through $\phi(s)$, and that in the feature space, the guide-policy $\pi^g$ cover the states visited by the optimal policy:

$$\sup_{s,h} \frac{d_h^{\pi^*}(\phi(s))}{d_h^{\pi^g}(\phi(s))} \leq C.$$

We provide formal definition of the marginalized state occupancy distribution in Appendix A.4. In other words, the guide-policy visits only all good *states in the feature space*. A policy that satisfies Assumption 4.2 may be far from optimal due to wrong choice of actions in each step. Assumption 4.2 is also much weaker than the *single policy concentratability coefficient* assumption, which requires the guide-policy visits all good *state and action* pairs and is a standard assumption in the literature in offline learning Rashidinejad et al. (2021); Xie et al. (2021). The ratio in Assumption 4.2 is also sometimes referred to as the *distribution mismatch coefficient* in the literature of policy gradient methods Agarwal et al. (2021).

We show via the following theorem that given Assumption 4.2, a simplified JSRL algorithm which only explores at current guide step $h + 1$ gives good performance guarantees for both tabular MDP and MDP with general function approximation. The simplified JSRL algorithm coincides with the Policy Search by Dynamic Programming (PSDP) algorithm in Bagnell et al. (2003), although our method is mainly motivated by the problem of fine-tuning and efficient exploration in value based methods, while PSDP focuses on policy-based methods.

**Theorem 4.3** (Informal)**.** *Under Assumption 4.2 and an appropriate choice of* TrainPolicy *and* EvaluatePolicy*, JSRL in Algorithm 1 guarantees a suboptimality of $O(CH^{5/2}S^{1/2}A/T^{1/2})$ for tabular MDP; and a near-optimal bound up to factor of $C \cdot \mathrm{poly}(H)$ for MDP with general function approximation.*

To achieve a polynomial bound for JSRL, it suffices to take TrainPolicy as $\epsilon$-greedy. This is in sharp contrast to Theorem 4.1, where $\epsilon$-greedy suffers from exponential sample complexity. As is discussed in the related work section, although polynomial and even near-optimal bound can be achieved by many optimism-based methods Jin et al. (2018); Ouyang et al. (2017), the JSRL algorithm does not require constructing a bonus function for uncertainty quantification, and can be implemented easily based on naïve $\epsilon$-greedy methods. Furthermore, although we focus on analyzing the simplified JSRL which only updates policy $\pi$ at current guide steps $h + 1$, in practice we run a JSRL algorithm as in Algorithm 1, which updates all policies after step $h + 1$. This is the main difference between our proposed algorithm and PSDP. For a formal statement and more discussion related to Theorem 4.3, please refer to Appendix A.4.3.

## 5 EXPERIMENTS

In our experimental evaluation, we study the following questions: (1) How does JSRL compare with competitive IL+RL baselines? (2) Does JSRL scale to complex vision-based robotic manipulation tasks? (3) How sensitive is JSRL to the quality of the guide-policy? (4) How important is the curriculum component of JSRL? (5) Does JSRL generalize? That is, can a guide-policy still be useful if it was pre-trained on a related task?

### 5.1 COMPARISON WITH IL+RL BASELINES

To study how JSRL compares with competitive IL+RL methods, we utilize the D4RL (Fu et al., 2020) benchmark tasks, which vary in task complexity and offline dataset quality. We focus on the most challenging D4RL tasks: Ant Maze and Adroit manipulation. We consider a common setting where the agent first trains on an offline dataset (1m transitions for Ant Maze, 100k transitions for Adroit) and then runs online fine-tuning for 1m steps. We compare against algorithms designed specifically for this setting, which include AWAC (Nair et al., 2020), IQL (Kostrikov et al., 2021), CQL (Kumar et al., 2020), and behavior cloning (BC). While JSRL can be used in combination with any initial guide-policy or fine-tuning algorithm, we show the combination of JSRL with the strongest baseline, IQL. IQL (Implicit Q-Learning) is an actor-critic method that completely avoids estimating the values of actions that are not seen in the offline dataset. This is a recent state-of-the-art method for the IL+RL setting we consider. In Table 1, we see that across the Ant Maze environments and Adroit environments, IQL+JSRL is able to successfully fine-tune given an initial offline dataset, and is competitive with baselines. We will come back for further analysis of Table 1 when discussing the sensitivity to the size of the dataset.

---

[1]The AWAC, BC, and CQL performance scores for D4RL are taken from Kostrikov et al. (2021) which only evaluated settings with full-sized datasets.

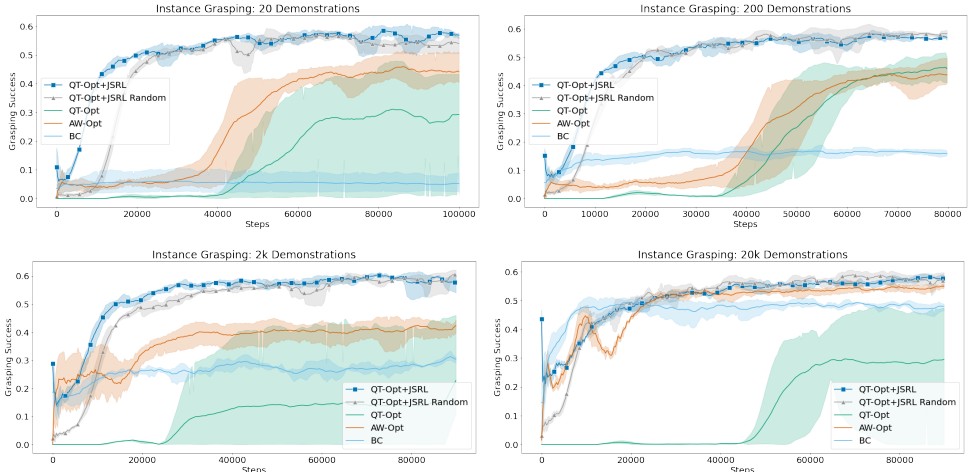

Figure 3: We evaluate the importance of guide-policy quality for JSRL on Instance Grasping, the most challenging task we consider. By limiting the initial demonstrations, JSRL is less sensitive to limitations of initial demonstrations compared to baselines, especially in the small-data regime. For each of these initial demonstration settings, we find that Qt-Opt+JSRL is more sample efficient than Qt-Opt+JSRL-Random in early stages of training, but converge to the same final performances. A similar analysis for Indiscriminate Grasping is provided in Fig. 10 in the Appendix.

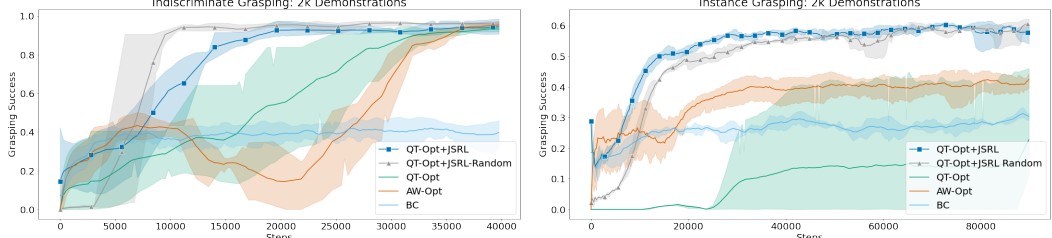

Figure 4: IL+RL methods on two simulated robotic grasping tasks. The baselines show improvement with fine-tuning, but Qt-Opt+JSRL is more sample efficient and attains higher final performance.

## 5.2 VISION-BASED ROBOTIC TASKS

Utilizing offline data is challenging in complex tasks such as vision-based robotic manipulation. The high dimensionality of both the continuous control action space as well as the pixel-based state space present unique scaling challenges for IL+RL methods. To study how JSRL scales to such settings, we focus on two simulated robotic manipulation tasks: Indiscriminate Grasping and Instance Grasping. In these tasks, a simulated robot arm is placed in front of a table with various categories of objects. When the robot lifts any object, a sparse reward is given for the Indiscriminate Grasping task; for the more challenging Instance Grasping task, the sparse reward is only given when a sampled target object is grasped. An image of the task is shown in Fig. 5 and described in detail in Appendix A.1.2. We compare JSRL against methods that have been shown to scale to such complex vision-based robotics settings: Qt-Opt (Kalashnikov et al., 2018), AW-Opt (Lu et al., 2021), and BC. Each method has access to the same offline dataset of 2,000 successful demonstrations and is allowed to run online fine-tuning for up to 100,000 steps. While AW-Opt and BC utilize offline successes as part of their original design motivation, we allow a more fair comparison for Qt-Opt by initializing the replay buffer with the offline demonstrations, which was not the case in the original Qt-Opt paper. Since we have already shown that JSRL can work well with an offline RL algorithm in the previous experiment, to demonstrate the flexibility of our approach, in this experiment we combine JSRL with an online Q-learning method: Qt-Opt. As seen in Fig. 4, the combination of Qt-Opt+JSRL (both versions of the curricula) outperforms the other methods in both sample efficiency as well as final performance.

## 5.3 INITIAL DATASET SENSITIVITY

While most IL+RL methods are improved by more data and higher quality data, there are often practical limitations that restrict initial offline datasets. JSRL is no exception to this dependency, as

| Environment | Dataset | AWAC[1] | BC[1] | CQL[1] | IQL | IQL+JSRL (Ours) | |
| --- | --- | --- | --- | --- | --- | --- | --- |
| | | | | | | Curriculum | Random |
| antmaze-umaze-v0 | 1k | – | – | – | $0.2 \pm 0.5$ | $\mathbf{15.6 \pm 19.9}$ | $10.4 \pm 9.6$ |
| | 10k | – | – | – | $55.5 \pm 12.5$ | $\mathbf{71.7 \pm 14.5}$ | $52.3 \pm 26.7$ |
| | 100k | – | – | – | $74.2 \pm 25.6$ | $\mathbf{93.7 \pm 4.2}$ | $92.1 \pm 2.8$ |
| | 1m (standard) | 59.0 | 54.6 | 99.4 | $97.6 \pm 3.2$ | $\mathbf{98.1 \pm 1.4}$ | $95.0 \pm 3.0$ |
| antmaze-umaze-diverse-v0 | 1k | – | – | – | $0.0 \pm 0.0$ | $3.1 \pm 8.0$ | $1.9 \pm 4.8$ |
| | 10k | – | – | – | $33.1 \pm 10.7$ | $\mathbf{72.6 \pm 12.2}$ | $39.4 \pm 20.1$ |
| | 100k | – | – | – | $29.9 \pm 23.1$ | $\mathbf{81.3 \pm 23.0}$ | $\mathbf{82.3 \pm 14.2}$ |
| | 1m (standard) | 49.0 | 45.6 | 99.4 | $53.0 \pm 30.5$ | $88.6 \pm 16.3$ | $89.8 \pm 10.0$ |
| antmaze-medium-play-v0 | 1k | – | – | – | $0.0 \pm 0.0$ | $0.0 \pm 0.0$ | $0.0 \pm 0.0$ |
| | 10k | – | – | – | $0.1 \pm 0.3$ | $\mathbf{16.7 \pm 12.9}$ | $3.8 \pm 5.0$ |
| | 100k | – | – | – | $32.8 \pm 32.6$ | $\mathbf{86.7 \pm 3.7}$ | $56.2 \pm 28.8$ |
| | 1m (standard) | 0.0 | 0.0 | 0.0 | $\mathbf{92.8 \pm 2.7}$ | $91.1 \pm 3.9$ | $87.8 \pm 4.2$ |
| antmaze-medium-diverse-v0 | 1k | – | – | – | $0.0 \pm 0.0$ | $0.0 \pm 0.0$ | $0.0 \pm 0.0$ |
| | 10k | – | – | – | $0.0 \pm 0.0$ | $\mathbf{16.6 \pm 11.7}$ | $5.1 \pm 8.2$ |
| | 100k | – | – | – | $15.7 \pm 17.7$ | $\mathbf{81.5 \pm 18.8}$ | $67.0 \pm 17.4$ |
| | 1m (standard) | 0.3 | 0.0 | 32.3 | $92.4 \pm 4.5$ | $\mathbf{93.1 \pm 3.1}$ | $86.3 \pm 5.9$ |
| antmaze-large-play-v0 | 1k | – | – | – | $0.0 \pm 0.0$ | $0.0 \pm 0.0$ | $0.0 \pm 0.0$ |
| | 10k | – | – | – | $0.0 \pm 0.0$ | $0.1 \pm 0.2$ | $0.0 \pm 0.0$ |
| | 100k | – | – | – | $2.6 \pm 8.2$ | $\mathbf{36.3 \pm 16.4}$ | $17.7 \pm 13.4$ |
| | 1m (standard) | 0.0 | 0.0 | 0.0 | $\mathbf{62.4 \pm 12.4}$ | $62.9 \pm 11.3$ | $48.6 \pm 10.0$ |
| antmaze-large-diverse-v0 | 1k | – | – | – | $0.0 \pm 0.0$ | $0.0 \pm 0.0$ | $0.0 \pm 0.0$ |
| | 10k | – | – | – | $0.0 \pm 0.0$ | $0.1 \pm 0.2$ | $0.0 \pm 0.0$ |
| | 100k | – | – | – | $4.1 \pm 10.4$ | $\mathbf{34.4 \pm 23.0}$ | $22.4 \pm 15.4$ |
| | 1m (standard) | 0.0 | 0.0 | 0.0 | $\mathbf{68.3 \pm 8.9}$ | $\mathbf{68.3 \pm 8.8}$ | $58.3 \pm 6.5$ |
| pen-binary-v0 | 100 | – | – | – | $18.8 \pm 11.6$ | $24.3 \pm 12.1$ | $\mathbf{29.1 \pm 7.6}$ |
| | 1k | – | – | – | $30.1 \pm 10.2$ | $36.7 \pm 7.9$ | $\mathbf{46.3 \pm 6.3}$ |
| | 10k | – | – | – | $38.4 \pm 11.2$ | $44.3 \pm 6.2$ | $\mathbf{52.1 \pm 3.3}$ |
| | 100k (standard) | $\mathbf{70.3}$ | 0.0 | 9.9 | $65.0 \pm 2.9$ | $62.6 \pm 3.6$ | $60.6 \pm 2.7$ |
| door-binary-v0 | 100 | – | – | – | $0.8 \pm 3.8$ | $0.4 \pm 1.8$ | $0.1 \pm 0.2$ |
| | 1k | – | – | – | $0.5 \pm 1.5$ | $0.7 \pm 1.0$ | $0.45 \pm 1.2$ |
| | 10k | – | – | – | $10.6 \pm 14.1$ | $4.3 \pm 8.4$ | $\mathbf{22.3 \pm 11.6}$ |
| | 100k (standard) | 30.1 | 0.0 | 0.0 | $\mathbf{50.2 \pm 2.5}$ | $28.5 \pm 19.5$ | $24.3 \pm 11.5$ |
| relocate-binary-v0 | 100 | – | – | – | $0.0 \pm 0.0$ | $\mathbf{0.0 \pm 0.1}$ | $0.0 \pm 0.0$ |
| | 1k | – | – | – | $0.0 \pm 0.0$ | $\mathbf{0.0 \pm 0.1}$ | $0.0 \pm 0.0$ |
| | 10k | – | – | – | $0.2 \pm 0.3$ | $\mathbf{0.6 \pm 1.6}$ | $0.5 \pm 0.7$ |
| | 100k (standard) | 2.7 | 0.0 | 0.0 | $\mathbf{8.6 \pm 7.7}$ | $0.0 \pm 0.1$ | $4.7 \pm 4.2$ |

Table 1: Comparing JSRL with IL+RL baselines on D4RL tasks by using averaged normalized scores for D4RL Ant Maze and Adroit tasks. Each method pretrains on an offline dataset and then runs online finetuning for 1m steps. Our method IQL+JSRL is competitive with IL+RL baselines in the full dataset setting, but performs significantly better in the small-data regime. For implementation details and more detailed comparisons, see Appendix A.2.

| Environment | # Demos | Qt-Opt | AW-Opt | BC | Qt-Opt+JSRL (Ours) |
| --- | --- | --- | --- | --- | --- |
| Indiscriminate Grasping | 20 | $0.0 \pm 0.0$ | $0.0 \pm 0.0$ | $0.19 \pm 0.04$ | $\mathbf{0.92 \pm 0.00}$ |
| Indiscriminate Grasping | 200 | $0.93 \pm 0.01$ | $\mathbf{0.96 \pm 0.02}$ | $0.23 \pm 0.00$ | $0.92 \pm 0.01$ |
| Indiscriminate Grasping | 2k | $0.94 \pm 0.01$ | $\mathbf{0.97 \pm 0.01}$ | $0.44 \pm 0.05$ | $0.94 \pm 0.03$ |
| Indiscriminate Grasping | 20k | $0.94 \pm 0.01$ | $\mathbf{0.98 \pm 0.01}$ | $0.91 \pm 0.01$ | $0.95 \pm 0.00$ |
| Instance Grasping | 20 | $0.23 \pm 0.20$ | $0.47 \pm 0.04$ | $0.05 \pm 0.04$ | $\mathbf{0.50 \pm 0.09}$ |
| Instance Grasping | 200 | $0.47 \pm 0.04$ | $0.49 \pm 0.02$ | $0.15 \pm 0.02$ | $\mathbf{0.54 \pm 0.03}$ |
| Instance Grasping | 2k | $0.15 \pm 0.26$ | $0.43 \pm 0.03$ | $0.28 \pm 0.04$ | $\mathbf{0.57 \pm 0.07}$ |
| Instance Grasping | 20k | $0.28 \pm 0.25$ | $0.57 \pm 0.01$ | $0.49 \pm 0.02$ | $\mathbf{0.58 \pm 0.02}$ |

Table 2: Limiting the initial number of demonstrations is challenging for IL+RL baselines on the difficult robotic grasping tasks. Notably, only Qt-Opt+JSRL is able to learn in the smallest-data regime of just 20 demonstrations, 100x less than the standard 2,000 demonstrations.

the quality of the guide-policy $\pi^g$ directly depends on the offline dataset when utilizing JSRL in an IL+RL setting (i.e., when the guide-policy is pre-trained on an offline dataset). We study the offline dataset sensitivity of IL+RL algorithms and JSRL on both D4RL tasks as well as the vision-based robotic grasping tasks. The two settings presented in D4RL and Robotic Grasping are quite different: IQL+JSRL in D4RL pretrains with an offline RL algorithm from a mixed quality offline dataset, while Qt-Opt+JSRL pretrains with BC from a high quality dataset.

For D4RL, methods typically use 1 million transitions from mixed-quality policies from previous RL training runs; as we reduce the size of the offline datasets in Table 1, IQL+JSRL performance degrades less than the baseline IQL performance. For the robotic grasping tasks, we provided 2,000 high-quality demonstrations. As we reduce the number of demonstrations, we find that JSRL efficiently

learns better policies. Across both D4RL and the robotic grasping tasks, JSRL outperforms baselines in the low-data regime, as shown in Table 1 and Table 2. In the high-data regime, when we increase the number of demonstrations by 10x to 20,000 demonstrations, we notice that AW-Opt and BC perform much more competitively, suggesting that the exploration challenge is no longer the bottleneck. While starting with such large numbers of demonstrations is not typically a realistic setting, this results suggests that the benefits of JSRL are most prominent when the offline dataset does not densely cover good state-action pairs. This aligns with our analysis in Appendix A.1 that JSRL does not require such assumptions about the dataset, but solely requires a prior policy.

### 5.4 JSRL-Curriculum vs. JSRL-Random Switching

In order to disentangle these two components, we propose an augmentation of our method, JSRL-Random, that randomly selects the number of guide-steps every episode. Using the D4RL tasks and the robotic grasping tasks, we compare JSRL-Random to JSRL and previous IL+RL baselines and find that JSRL-Random performs quite competitively, as seen in Table 1 and Table 2. However, when considering sample efficiency, Fig. 4 shows that JSRL is better than JSRL-Random in early stages of training, while converged performance is comparable. These same trends hold when we limit the quality of the guide-policy by constraining the initial dataset, as seen in Fig. 3. This suggests that while a curriculum of guide-steps does help sample efficiency, the largest benefits of JSRL may stem from the *presence* of good visitation states induced by the guide-policy as opposed to the specific *order* of good visitation states, as suggested by our analysis in Appendix A.4.3. For analyze hyperparameter sensitivity of JSRL-Curriculum and provide the specific implementation of hyperparameters chosen for our experiments in Appendix A.3.

### 5.5 Guide-Policy Generalization

In order to study how guide-policies from easier tasks can be used to efficiently explore more difficult tasks, we train an indiscriminate grasping policy and use it as the guide-policy for JSRL on instance grasping (Figure 13). While the performance when using the indiscriminate guide is worse than using the instance guide, the performance for both JSRL versions outperform vanilla Qt-Opt. We also test JSRL 's generalization capabilities in the D4RL setting. We consider two variations of Ant mazes: "play" and "diverse". In antmaze-*-play, the agent must reach a fixed set of goal locations from a fixed set of starting locations. In antmaze-*-diverse, the agent must reach random goal locations from random starting locations. Thus, the diverse environments present a greater challenge than the corresponding play environments. In Figure 14, we see that JSRL is able to better generalize to unseen goal and starting locations compared to vanilla IQL.

## 6 Conclusion

In this work, we propose Jump-Start Reinforcement Learning (JSRL), a method for leveraging a prior policy of any form to bolster exploration in RL to increase sample efficiency. Our algorithm creates a learning curriculum by rolling in a pre-existing guide-policy, which is then followed by the self-improving exploration policy. The job of the exploration-policy is simplified, as it starts its exploration from states closer to the goal. As the exploration policy improves, the effect of the guide-policy diminishes, leading to a fully capable RL policy. Importantly, our approach is generic since it can be used with any RL method including value-based RL approaches, which have traditionally struggled in this setting. We showed the benefits of JSRL in a set of offline RL benchmark tasks as well as more challenging vision-based robotic simulation tasks. Our experiments indicate that JSRL is more sample efficient than more complex IL+RL approaches while being compatible with other approaches' benefits. In addition, we presented theoretical analysis of an upper bound on the sample complexity of JSRL , which showed from-exponential-to-polynomial improvement in time horizon from non-optimism exploration methods. In the future, we plan on deploying JSRL in the real world in conjunction with various types of guide-policies to further investigate its ability to bootstrap data efficient RL.

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

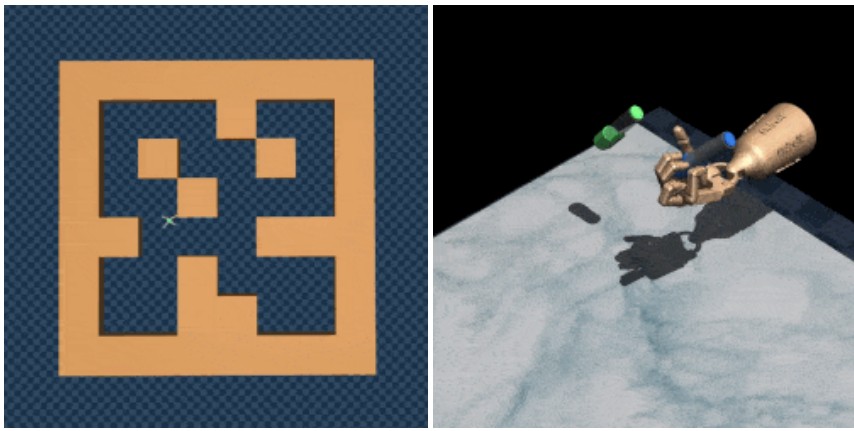

Figure 6: Example ant maze (left) and adroit dexterous manipulation (right) tasks.

# A APPENDIX

## A.1 EXPERIMENT IMPLEMENTATION DETAILS

### A.1.1 D4RL: ANT MAZE AND ADROIT

We evaluate on the Ant Maze and Adroit tasks, the most challenging tasks in the D4RL benchmark Fu et al. (2020). For the baseline IL+RL method comparisons, we utilize implementations and reported results from Kostrikov et al. (2021): we use the open-sourced version of IQL and the reported results from for AWAC, BC, and CQL. While the standard initial offline datasets contain 1m transitions for Ant Maze and 100k transitions for Adroit, we additionally ablate the datasets to evaluate settings with 100, 1k, 10k, and 100k transitions provided initially.

For the implementation of IQL+JSRL, we build upon the open-sourced IQL implementation Kostrikov et al. (2021). First, to obtain a guide-policy, we use IQL without modification for pretraining on the offline dataset. Then, we follow Algorithm 1 when finetuning online and use the IQL online update as the TRAINPOLICY step from Algorithm 1. The IQL neural network architecture follows the original implementation of Kostrikov et al. (2021). For finetuning, we maintain two replay buffers for offline and online transitions. The offline buffer contains all the demonstrations, and the online buffer is FIFO with a fixed capacity of 100k transitions. For each gradient update during finetuning, we sample minibatches such that 75% of samples come from the online buffer, and 25% of samples come from the offline buffer.

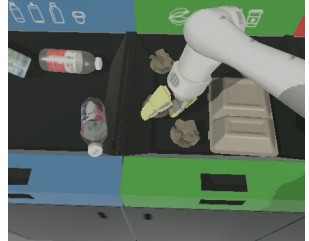

Figure 5: In the simulated vision-based robotic grasping tasks, a robot arm must grasp various objects placed in bins in front of it. Full implementation details are described in Appendix A.1.2.

Our implementation of IQL+JSRL focused on two settings when switching from offline pretraining to online finetuning: Warm-starting and Cold-starting. When Warm-starting, we copy the actor, critic, target critic, and value networks from the pre-trained guide-policy to the exploration-policy. When Cold-starting, we instead start training the exploration-policy from scratch. Results for both variants are shown in Appendix A.2. We find that empirically, the performance of these two variants is highly dependent on task difficulty as well as the quality of the initial offline dataset. When initial datasets are very poor, cold-starting usually performs better; when initial datasets are dense and high-quality, warm-starting seems to perform better. For the results reported in Table 1, we utilize Cold-start results for both IQL+JSRL-Curriculum and IQL+JSRL-Random.

Finally, the curriculum implementation for IQL+JSRL used policy evaluation every 10,000 steps to gauge learning progress of the exploration-policy $\pi^e$. When the moving average of $\pi^e$'s performance

increases over a few samples, we move on to the next curriculum stage. For the IQL+JSRL-Random variant, we randomly sample the number of guide-steps for every single episode.

### A.1.2 SIMULATED ROBOTIC MANIPULATION

We simulate a 7 DoF arm with an over-the-shoulder camera (see Figure 5) Three bins in front of the robot are filled with various simulated objects to be picked up by the robot and a sparse binary reward is assigned if any object is lifted above a bin at the end of an episode. States are represented in the form of RGB images and actions are continuous Cartesian displacements of the gripper's 3D positions and yaw. In addition, the policy commands discrete gripper open and close actions and may terminate an episode.

For the implementation of Qt-Opt+JSRL, we build upon the Qt-Opt algorithm described in Kalashnikov et al. (2018). First, to obtain a guide-policy we use a BC policy trained offline on the provided demonstrations. Then, we follow Algorithm 1 when finetuning online and use the Qt-Opt online update as the TRAINPOLICY step from Algorithm 1. The demonstrations are not added to the Qt-Opt+JSRLreplay buffer. The Qt-Opt neural network architecture follows the original implementation in Kalashnikov et al. (2018).

Finally, similar to Appendix A.1.1, the curriculum implementation for Qt-Opt+JSRLused policy evaluation every 1,000 steps to gauge learning progress of the exploration-policy $\pi^e$. When the moving average of $\pi^e$'s performance increases over a few samples, the number of guide-steps is lowered, allowing the JSRL curriculum to continue. For the Qt-Opt+JSRL-Random variant, we randomly sample the number of guide-steps for every single episode.

### A.2 ADDITIONAL EXPERIMENTS

| Environment | JSRL: Random Switching | | JSRL: Curriculum | | IQL |
|---|---|---|---|---|---|
| | Warm-start | Cold-start | Warm-start | Cold-start | |
| pen-binary-v0 | $27.18 \pm 7.77$ | $\mathbf{29.12 \pm 7.62}$ | $25.10 \pm 8.73$ | $24.31 \pm 12.05$ | $18.80 \pm 11.63$ |
| door-binary-v0 | $0.01 \pm 0.04$ | $0.06 \pm 0.23$ | $\mathbf{1.45 \pm 4.67}$ | $0.40 \pm 1.80$ | $0.84 \pm 3.76$ |
| relocate-binary-v0 | $0.00 \pm 0.00$ | $0.00 \pm 0.00$ | $0.00 \pm 0.00$ | $\mathbf{0.01 \pm 0.06}$ | $0.01 \pm 0.03$ |

Table 3: Adroit 100 Offline Transitions

| Environment | JSRL: Random Switching | | JSRL: Curriculum | | IQL |
|---|---|---|---|---|---|
| | Warm-start | Cold-start | Warm-start | Cold-start | |
| pen-binary-v0 | $\mathbf{47.23 \pm 3.96}$ | $46.30 \pm 6.34$ | $34.23 \pm 7.22$ | $36.74 \pm 7.91$ | $30.11 \pm 10.22$ |
| door-binary-v0 | $0.15 \pm 0.25$ | $0.45 \pm 1.22$ | $0.44 \pm 0.89$ | $\mathbf{0.68 \pm 1.02}$ | $0.53 \pm 1.46$ |
| relocate-binary-v0 | $\mathbf{0.06 \pm 0.08}$ | $0.01 \pm 0.04$ | $0.05 \pm 0.09$ | $0.04 \pm 0.10$ | $0.01 \pm 0.03$ |

Table 4: Adroit 1k Offline Transitions

| Environment | IQL+JSRL: Random Switching | | IQL+JSRL: Curriculum | | IQL |
|---|---|---|---|---|---|
| | Warm-start | Cold-start | Warm-start | Cold-start | |
| pen-binary-v0 | $51.78 \pm 3.00$ | $\mathbf{52.11 \pm 3.30}$ | $38.04 \pm 12.71$ | $44.31 \pm 6.22$ | $38.41 \pm 11.18$ |
| door-binary-v0 | $10.59 \pm 11.78$ | $\mathbf{22.32 \pm 11.61}$ | $5.08 \pm 7.60$ | $4.33 \pm 8.38$ | $10.61 \pm 14.11$ |
| relocate-binary-v0 | $1.99 \pm 3.15$ | $0.50 \pm 0.65$ | $\mathbf{4.39 \pm 8.17}$ | $0.55 \pm 1.60$ | $0.19 \pm 0.32$ |

Table 5: Adroit 10k Offline Transitions

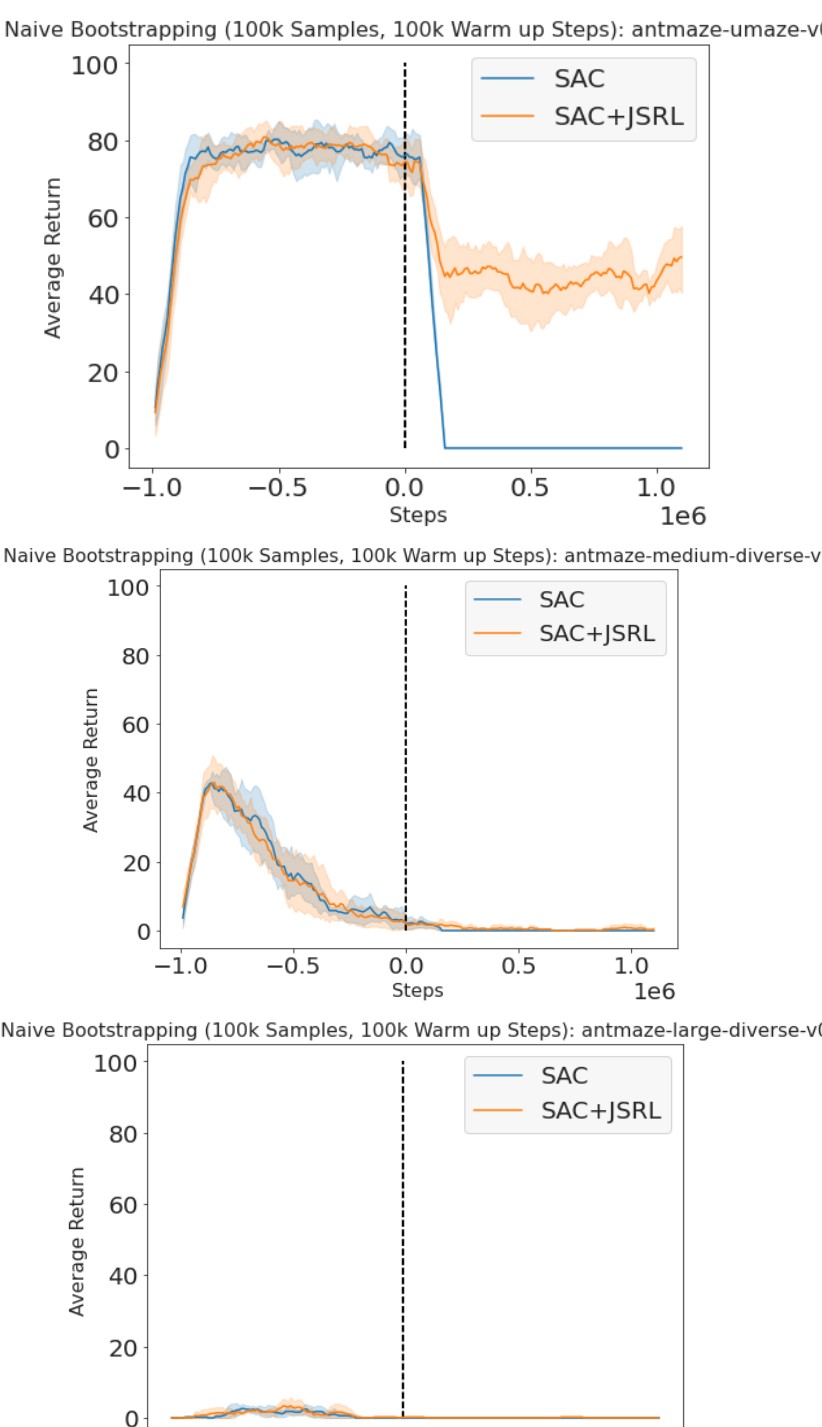

Figure 7: A policy is first pre-trained on 100k offline transitions. Negative steps correspond to this pre-training. We then roll out the pre-trained policy for 100k timesteps, and use these online samples to warm-up the critic network. After warming up the critic, we continue with actor-critic fine-tuning with the pre-trained policy and the warmed up critic.

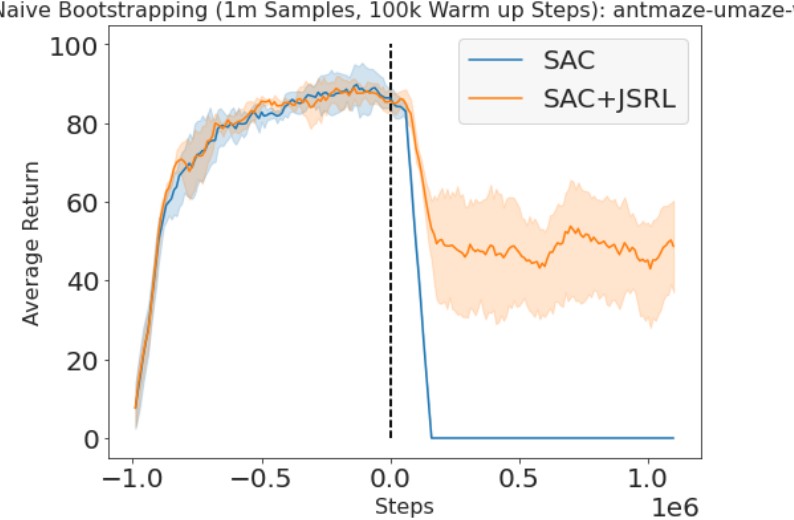

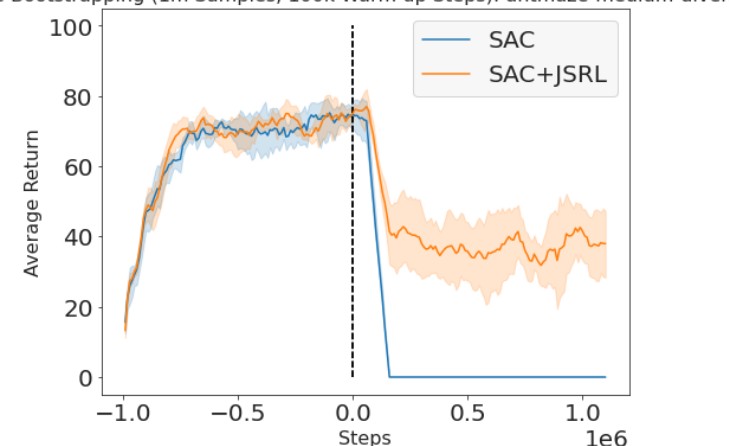

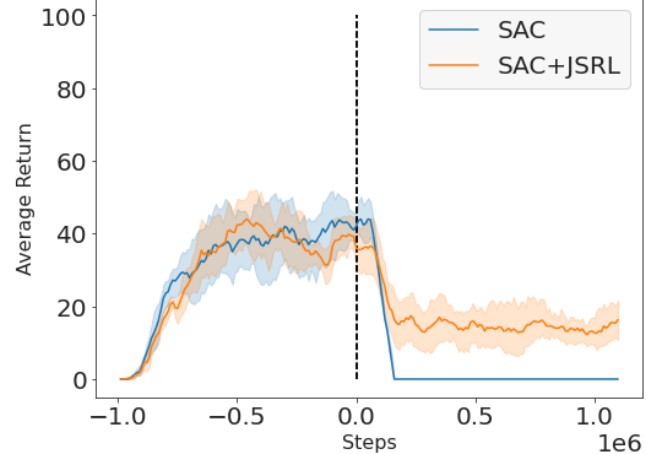

Figure 8: A policy is first pre-trained on one million offline transitions. Negative steps correspond to this pre-training. We then roll out the pre-trained policy for 100k timesteps, and use these online samples to warm-up the critic network. After warming up the critic, we continue with actor-critic fine-tuning with the pre-trained policy and the warmed up critic. Allowing the critic to warm up provides a stronger baseline to compare JSRL to, since in the case where we have a policy, but no value function, we could use that policy to train a value function.

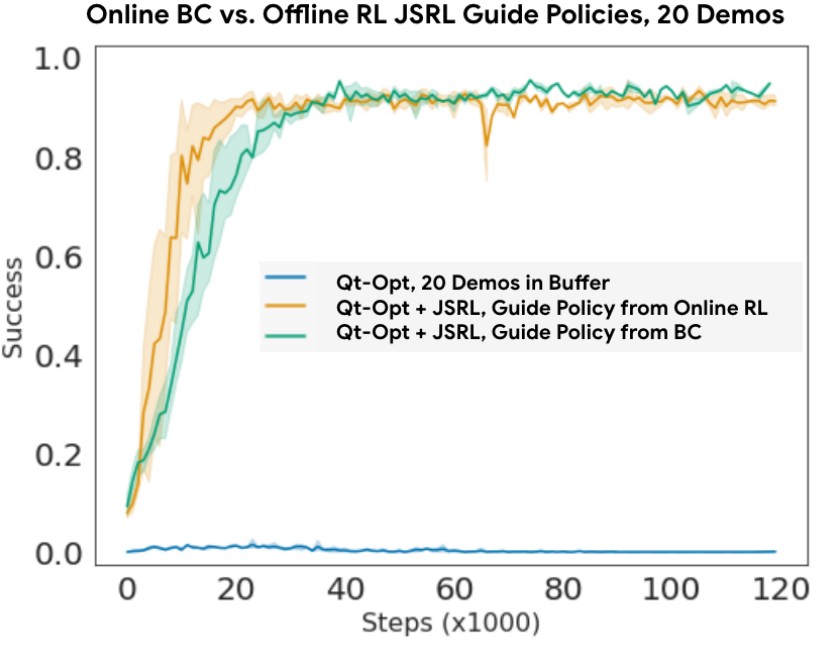

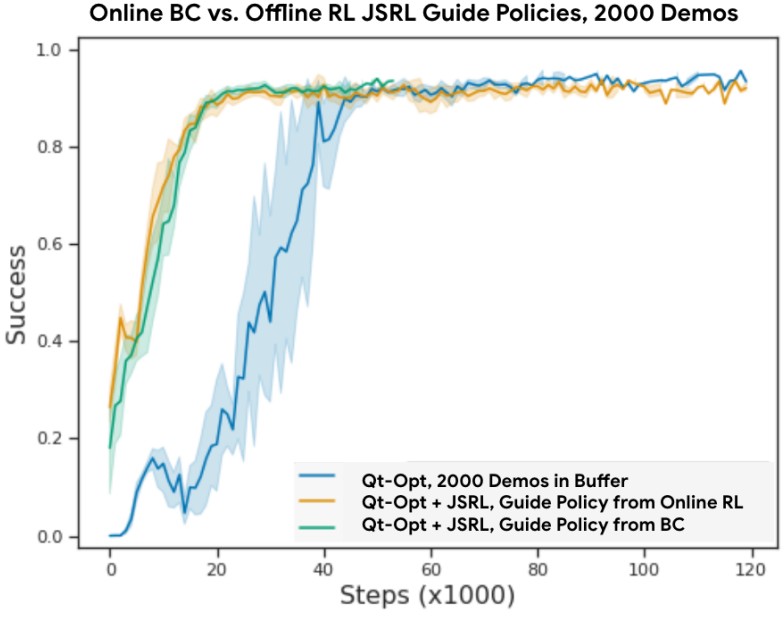

Figure 9: QT-Opt+JSRL using guide-policies trained from-scratch online vs. guide-policies trained with BC on demonstration data in the indiscriminate grasping environment. For each experiment, the guide-policy trained offline and the guide-policy trained online are of equivalent performance.

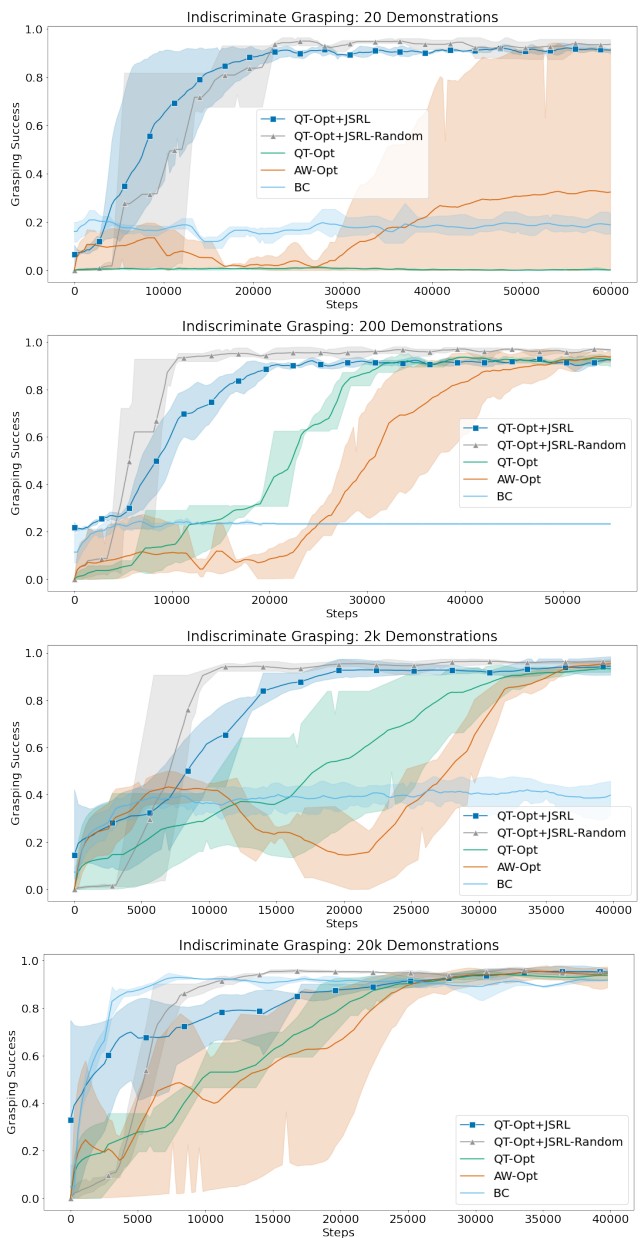

Figure 10: Comparing IL+RL methods with JSRL on the Indiscriminate Grasping task while adjusting the initial demonstrations available. In addition, compare the sample efficiency

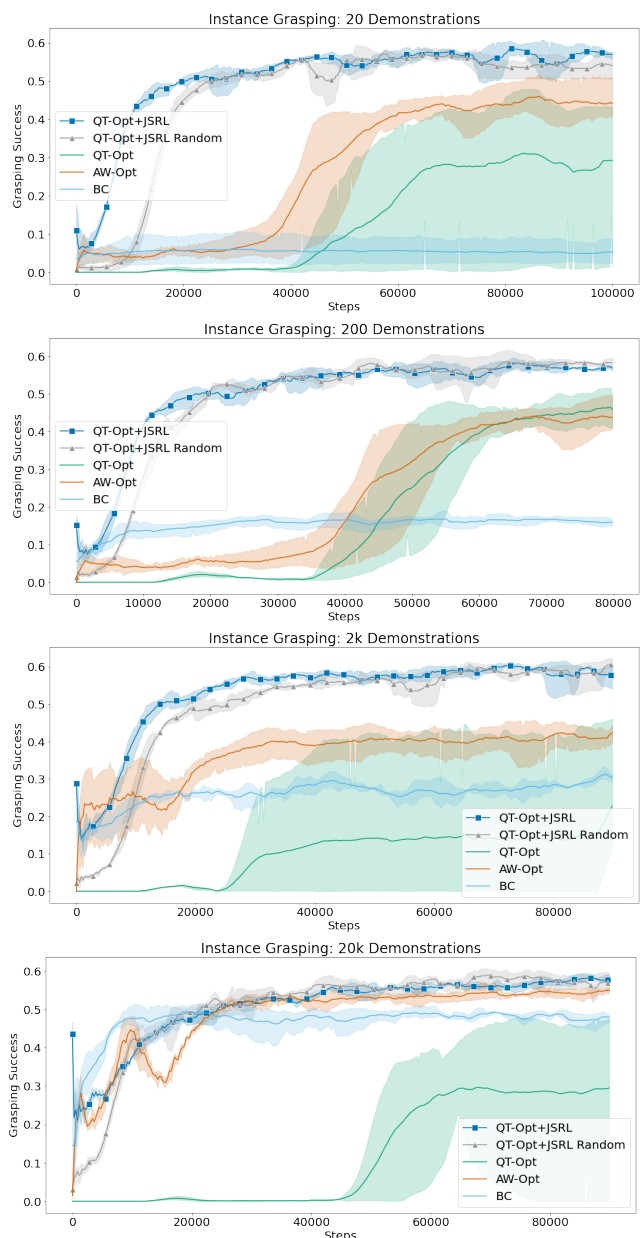

Figure 11: Comparing IL+RL methods with JSRL on the Instance Grasping task while adjusting the initial demonstrations available.

| Environment | IQL+JSRL: Random Switching | | IQL+JSRL: Curriculum | | IQL |
| | Warm-start | Cold-start | Warm-start | Cold-start | |
|---|---|---|---|---|---|
| pen-binary-v0 | $60.06 \pm 2.94$ | $60.58 \pm 2.73$ | $62.81 \pm 2.79$ | $62.59 \pm 3.62$ | $\mathbf{64.96 \pm 2.87}$ |
| door-binary-v0 | $27.23 \pm 8.90$ | $24.27 \pm 11.47$ | $38.70 \pm 17.25$ | $28.51 \pm 19.54$ | $\mathbf{50.21 \pm 2.50}$ |
| relocate-binary-v0 | $5.09 \pm 4.39$ | $4.69 \pm 4.16$ | $11.18 \pm 11.69$ | $0.04 \pm 0.14$ | $\mathbf{8.59 \pm 7.70}$ |

Table 6: Adroit 100k Offline Transitions

| Environment | IQL+JSRL: Random Switching | | IQL+JSRL: Curriculum | | IQL |
| | Warm-start | Cold-start | Warm-start | Cold-start | |
|---|---|---|---|---|---|
| antmaze-umaze-v0 | $0.10 \pm 0.31$ | $10.35 \pm 9.59$ | $0.40 \pm 0.94$ | $\mathbf{15.60 \pm 19.87}$ | $0.20 \pm 0.52$ |
| antmaze-umaze-diverse-v0 | $0.10 \pm 0.31$ | $1.90 \pm 4.81$ | $0.45 \pm 1.23$ | $\mathbf{3.05 \pm 7.99}$ | $0.00 \pm 0.00$ |
| antmaze-medium-play-v0 | $0.00 \pm 0.00$ | $0.00 \pm 0.00$ | $0.00 \pm 0.00$ | $0.00 \pm 0.00$ | $0.00 \pm 0.00$ |
| antmaze-medium-diverse-v0 | $0.00 \pm 0.00$ | $0.00 \pm 0.00$ | $0.00 \pm 0.00$ | $0.00 \pm 0.00$ | $0.00 \pm 0.00$ |
| antmaze-large-play-v0 | $0.00 \pm 0.00$ | $0.00 \pm 0.00$ | $0.00 \pm 0.00$ | $0.00 \pm 0.00$ | $0.00 \pm 0.00$ |
| antmaze-large-diverse-v0 | $0.00 \pm 0.00$ | $0.00 \pm 0.00$ | $0.00 \pm 0.00$ | $0.00 \pm 0.00$ | $0.00 \pm 0.00$ |

Table 7: Ant Maze 1k Offline Transitions

| Environment | IQL+JSRL: Random Switching | | IQL+JSRL: Curriculum | | IQL |
| | Warm-start | Cold-start | Warm-start | Cold-start | |
|---|---|---|---|---|---|
| antmaze-umaze-v0 | $56.00 \pm 13.70$ | $52.70 \pm 26.71$ | $57.25 \pm 15.86$ | $\mathbf{71.70 \pm 14.49}$ | $55.50 \pm 12.51$ |
| antmaze-umaze-diverse-v0 | $23.05 \pm 10.96$ | $39.35 \pm 20.07$ | $26.80 \pm 12.03$ | $\mathbf{72.55 \pm 12.18}$ | $33.10 \pm 10.74$ |
| antmaze-medium-play-v0 | $0.05 \pm 0.22$ | $3.75 \pm 4.97$ | $0.00 \pm 0.00$ | $\mathbf{16.65 \pm 12.93}$ | $0.10 \pm 0.31$ |
| antmaze-medium-diverse-v0 | $0.00 \pm 0.00$ | $5.10 \pm 8.16$ | $0.00 \pm 0.00$ | $\mathbf{16.60 \pm 11.71}$ | $0.00 \pm 0.00$ |
| antmaze-large-play-v0 | $0.00 \pm 0.00$ | $0.00 \pm 0.00$ | $0.00 \pm 0.00$ | $\mathbf{0.05 \pm 0.22}$ | $0.00 \pm 0.00$ |
| antmaze-large-diverse-v0 | $0.00 \pm 0.00$ | $0.00 \pm 0.00$ | $0.00 \pm 0.00$ | $\mathbf{0.05 \pm 0.22}$ | $0.00 \pm 0.00$ |

Table 8: Ant Maze 10k Offline Transitions

| Environment | IQL+JSRL: Random Switching | | IQL+JSRL: Curriculum | | IQL |
| | Warm-start | Cold-start | Warm-start | Cold-start | |
|---|---|---|---|---|---|
| antmaze-umaze-v0 | $73.35 \pm 22.58$ | $92.05 \pm 2.76$ | $71.35 \pm 26.36$ | $\mathbf{93.65 \pm 4.21}$ | $74.15 \pm 25.62$ |
| antmaze-umaze-diverse-v0 | $40.95 \pm 13.34$ | $\mathbf{82.25 \pm 14.20}$ | $38.80 \pm 21.96$ | $81.30 \pm 23.04$ | $29.85 \pm 23.08$ |
| antmaze-medium-play-v0 | $9.55 \pm 14.42$ | $56.15 \pm 28.78$ | $22.15 \pm 29.82$ | $\mathbf{86.85 \pm 3.67}$ | $32.80 \pm 32.64$ |
| antmaze-medium-diverse-v0 | $14.05 \pm 13.30$ | $67.00 \pm 17.43$ | $15.75 \pm 16.48$ | $\mathbf{81.50 \pm 18.80}$ | $15.70 \pm 17.69$ |
| antmaze-large-play-v0 | $0.35 \pm 0.93$ | $17.70 \pm 13.35$ | $0.45 \pm 1.19$ | $\mathbf{36.30 \pm 16.41}$ | $2.55 \pm 8.19$ |
| antmaze-large-diverse-v0 | $1.25 \pm 2.31$ | $22.40 \pm 15.44$ | $0.75 \pm 1.16$ | $\mathbf{34.35 \pm 22.97}$ | $4.10 \pm 10.37$ |

Table 9: Ant Maze 100k Offline Transitions

| Environment | IQL+JSRL: Random Switching | | IQL+JSRL: Curriculum | | IQL |
| | Warm-start | Cold-start | Warm-start | Cold-start | |
|---|---|---|---|---|---|
| antmaze-umaze-v0 | $95.35 \pm 2.23$ | $94.95 \pm 2.95$ | $96.70 \pm 1.69$ | $\mathbf{98.05 \pm 1.43}$ | $97.60 \pm 3.19$ |
| antmaze-umaze-diverse-v0 | $65.95 \pm 27.00$ | $\mathbf{89.80 \pm 10.00}$ | $59.95 \pm 33.90$ | $88.55 \pm 16.37$ | $52.95 \pm 30.48$ |
| antmaze-medium-play-v0 | $82.25 \pm 4.88$ | $87.80 \pm 4.20$ | $92.20 \pm 2.84$ | $91.05 \pm 3.86$ | $\mathbf{92.75 \pm 2.73}$ |
| antmaze-medium-diverse-v0 | $83.45 \pm 4.64$ | $86.25 \pm 5.94$ | $91.65 \pm 2.98$ | $\mathbf{93.05 \pm 3.10}$ | $92.40 \pm 4.50$ |
| antmaze-large-play-v0 | $50.35 \pm 9.74$ | $48.60 \pm 10.01$ | $\mathbf{72.15 \pm 9.66}$ | $62.85 \pm 11.31$ | $62.35 \pm 12.42$ |
| antmaze-large-diverse-v0 | $56.80 \pm 9.15$ | $58.30 \pm 6.54$ | $\mathbf{70.55 \pm 17.43}$ | $68.25 \pm 8.76$ | $68.25 \pm 8.85$ |

Table 10: Ant Maze 1m Offline Transitions

## A.3 HYPERPARAMETERS OF JSRL

JSRL introduces three hyperparameters: (1) the initial number of guide-steps that the guide-policy takes at the beginning of fine-tuning ($H_1$), (2) the number of curriculum stages ($n$), and (3) the

| | | Moving Average Horizon | | |
|---|---|---|---|---|
| | | 1 | 5 | 10 |
| | 0% | 79.66 | 56.66 | 74.83 |
| Tolerance | 5% | 51.12 | 78.8 | 79.78 |
| | 15% | 56.41 | 47.46 | 59.52 |

Table 11: We fix the number of curriculum stages at $n = 10$ for antmaze-large-diverse-v0, then vary the moving average horizon and tolerance. Each number is the average reward after 5 million training steps of one seed. As tolerance increases, the reward decreases since curriculum stages are not fully mastered before moving on.

performance threshold that decides whether to move on to the next curriculum stage ($\beta$). Minimal tuning was done for these hyperparameters.

**IQL+JSRL:** For offline pre-training and online fine-tuning, we use the same exact hyperparameters as the default implementation of IQL [6].

Our reported results for vanilla IQL do differ from the original paper, but this is due to us running more random seeds (20 vs. 5), which we also consulted with the authors of IQL. For Indiscriminate and Instance Grasping experiments we utilize the same environment, task definition, and training hyperparameters as Qt-Opt and AW-Opt.

**Initial Number of Guide-Steps:** $H_1$**:**

For all X+JSRLexperiments, we train the guide-policy (IQL for D4RL and BC for grasping) then evaluate it to determine how many steps it takes to solve the task on average. For D4RL, we evaluate it over one hundred episodes. For grasping, we plot training metrics and observe the average episode length after convergence. This average is then used as the initial number of guide-steps. Since $H_1$ is directly computed, no hyperparameter search is required.

**Curriculum Stages:** $n$

Once the number of curriculum stages was chosen, we computed the number of steps between curriculum stages as $\frac{H_1}{n}$. Then $h$ varies from $H_1 - \frac{H_1}{n}, H_1 - 2\frac{H_1}{n}, \ldots, H_1 - (n-1)\frac{H_1}{n}, 0$. To decide on an appropriate number of curriculum stages, we decreased $n$ (increased $\frac{H_1}{n}$ and $H_i - H_{i-1}$), starting from $n = H$, until the curriculum became too difficult for the agent to overcome (i.e., the agent becomes "stuck" on a curriculum stage). We then used the minimal value of $n$ for which the agent could still solve all stages. *In practice, we did not try every value between $H$ and $1$, but chose a very small subset of values to test in this range.*

**Performance Threshold** $\beta$**:** For both grasping and D4RL tasks, we evaluated $\pi$ between fixed intervals and computed the moving average of these evaluations (5 for D4RL, 3 for grasping). If the current moving average is close enough to the best previous moving average, then we move from curriculum stage $i$ to $i + 1$. To define "close enough", we set a tolerance that let the agent move to the next stage if the current moving average was within some percentage of the previous best. The tolerance and moving average horizon were our "$\beta$", a generic parameter that is flexible based on how costly it is to evaluate the performance of $\pi$. In Figure 12 and Table 11, we perform small studies to determine how varying $\beta$ affects JSRL's performance.

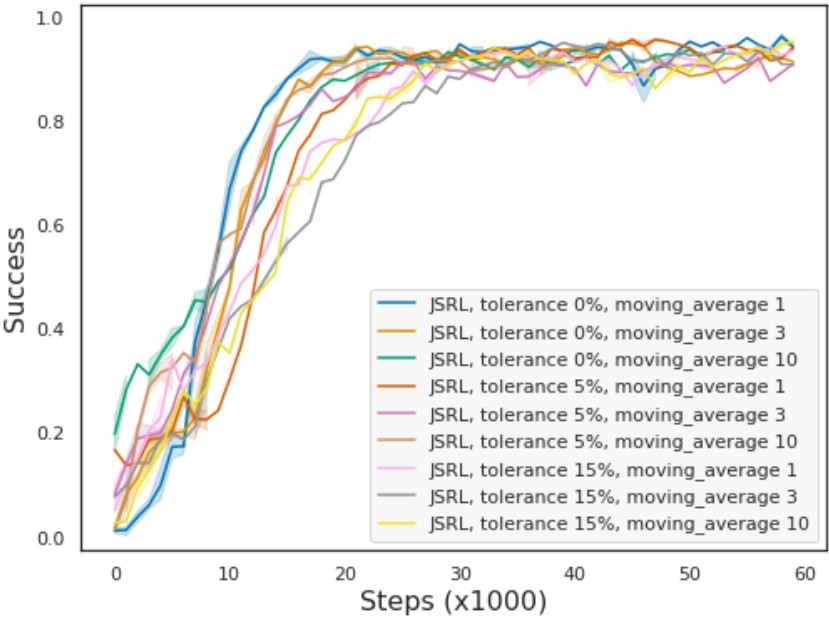

Figure 12: Ablation study for $\beta$ in the indiscriminate grasping environment. We find that the moving average horizon does not have a large impact on performance, but larger tolerance slightly hurts performance. A larger tolerance around the best moving average makes it easier for JSRL to move on to the next curriculum stage. This means that experiments with a larger tolerance could potentially move on to the next curriculum stage before JSRL masters the previous curriculum stage, leading to lower performance.

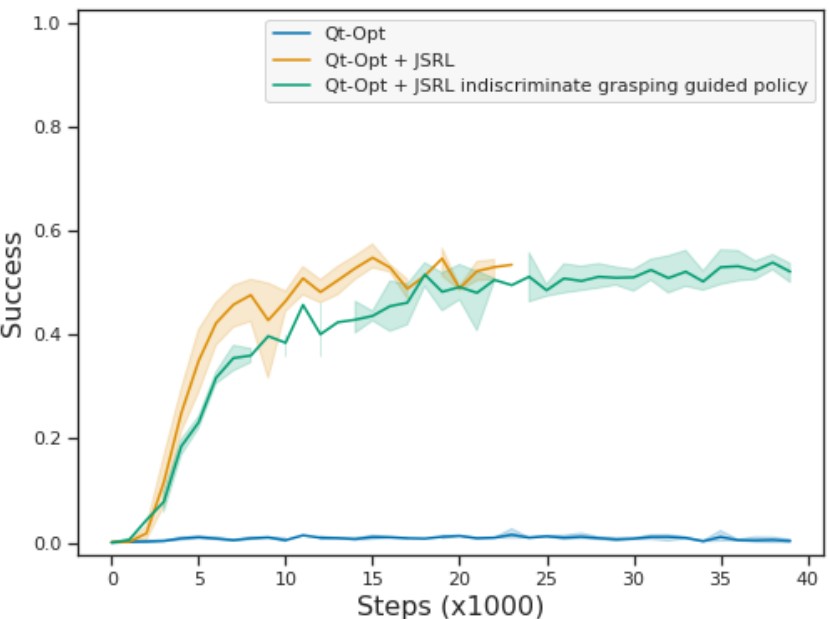

Figure 13: First, an indiscriminate grasping policy is trained using online QT-Opt to 90% indiscriminate grasping success and 5% instance grasping success (when the policy happens to randomly pick the correct object). We compare this 90% indiscriminate grasping guide policy with a 8.4% success instance grasping guide policy trained with BC on 2k demonstrations. While the performance for using the indiscriminate guide is slightly worse than using the instance guide, the performance for both JSRL versions are much better than vanilla Qt-Opt.

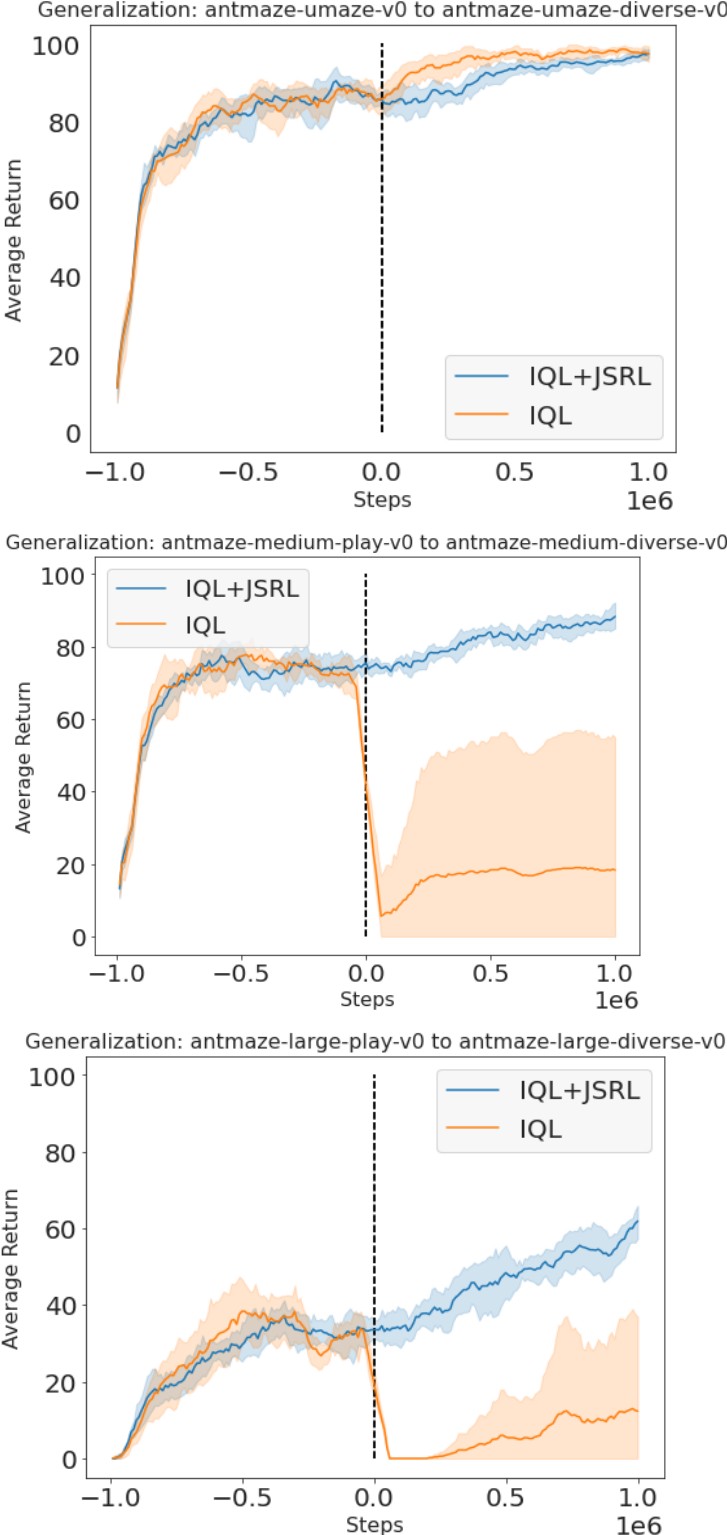

Figure 14: First, a policy is trained offline on a simpler antmaze-*-play environment for one million steps (depicted by negative steps). This policy is then used for initializing fine-tuning (depicted by positive steps) in a more complex antmaze-*-diverse environment. We find that IQL+JSRL can better generalize to the more difficult antmazes compared to IQL even when using guide-policies trained on different tasks.

### A.4 THEORETICAL ANALYSIS FOR JSRL

#### A.4.1 SETUP AND NOTATIONS

Consider a finite-horizon time-inhomogeneous MDP with a fixed total horizon $H$ and bounded reward $r_h \in [0, 1], \forall h \in [H]$. The transition of state-action pair $(s, a)$ in step $h$ is denoted as $\mathbb{P}_h(\cdot \mid s, a)$. Assume that at step $0$, the initial state follows a distribution $p_0$.

For simplicity, we use $\pi$ to denote the policy for $H$ steps $\pi = \{\pi_h\}_{h=1}^H$. We let $d_h^\pi(s)$ be the marginalized state occupancy distribution in step $h$ when we follow policy $\pi$.

#### A.4.2 PROOF SKETCH FOR THEOREM 4.1

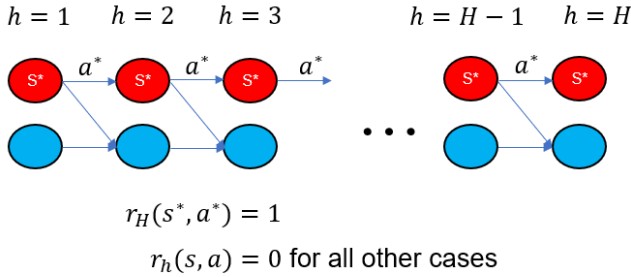

$$r_H(s^*, a^*) = 1$$
$$r_h(s, a) = 0 \text{ for all other cases}$$

Figure 15: Lower bound instance: combination lock

We construct a special instance, combination lock MDP, which is depicted in Figure 15 and works as follows. The agent can only arrive at the red state $s_{h+1}^\star$ in step $h + 1$ when it takes action $a_h^\star$ at the red state $s_h^\star$ at step $h$. Once it leaves state $s_h^\star$, the agent stays in the blue states and can never get back to red states again. At the last layer, one receives reward $1$ when the agent is at state $s_H^\star$ and takes action $a_H^\star$. For all other cases, the reward is $0$. In exploration from scratch, before seeing $r_H(s^\star, a^\star)$, one only sees reward $0$. Thus $0$-initialized $\epsilon$-greedy always takes each action with probability $1/2$. The probability of arriving at state $s_H^\star$ with uniform actions is $1/2^H$, which means that one needs at least $2^H$ samples in expectation to see $r_H(s^\star, a^\star)$.

#### A.4.3 UPPER BOUND OF JSRL

In this section, we restate Theorem 4.3 and its assumption in a formal way. First, we make assumption on the quality of the guide-policy, which is the key assumption that helps improve the exploration from exponential to polynomial sample complexity. One of the weakest assumption in theory of offline learning literature is the single policy concentratability coefficient Rashidinejad et al. (2021); Xie et al. (2021)[1]. Concretely, they assume that there exists a guide-policy $\pi^g$ such that

$$\sup_{s,a,h} \frac{d_h^{\pi^\star}(s, a)}{d_h^{\pi^g}(s, a)} \le C. \tag{1}$$

This means that for any state action pair that the optimal policy visits, the guide-policy shall also visit with certain probability.

In the analysis, we impose a strictly weaker assumption. We only require that the guide-policy visits all good *states in the feature space* instead of all good *state and action pairs*.

**Assumption A.1** (Quality of guide-policy $\pi^g$). Assume that the state is parametrized by some feature mapping $\phi : \mathcal{S} \to \mathbb{R}^d$ such that for any policy $\pi$, $Q^\pi(s, a)$ and $\pi(s)$ depends on $s$ only through $\phi(s)$. We assume that in the feature space, the guide-policy $\pi^g$ cover the states visited by the optimal policy:

$$\sup_{s,h} \frac{d_h^{\pi^\star}(\phi(s))}{d_h^{\pi^g}(\phi(s))} \le C.$$

---

[1]The single policy concentratability assumption is already a weaker version of the traditional concentratability coefficient assumption, which takes a supremum of the density ratio over all state-action pairs and all policies (Scherrer, 2014; Chen & Jiang, 2019; Jiang, 2019; Wang et al., 2019; Liao et al., 2020; Liu et al., 2019; Zhang et al., 2020a).

Note that for the tabular case when $\phi(s) = s$, one can easily prove that equation 1 implies Assumption A.1. In real robotics, the assumption implies that the guide-policy at least sees the features of the good states that the optimal policy also see. However, the guide-policy can be arbitrarily bad in terms of choosing actions.

Before we proceed to the main theorem, we need to impose another assumption on the performance of the exploration step, which requires to find an exploration algorithm that performs well in the case of $H = 1$ (contextual bandit).

**Assumption A.2** (Performance guarantee for ExplorationOracle_CB)**.** In (online) contextual bandit with stochastic context $s \sim p_0$ and stochastic reward $r(s, a)$ supported on $[0, R]$, there exists some ExplorationOracle_CB which executes a policy $\pi^t$ in each round $t \in [T]$, such that the total regret is bounded:

$$\sum_{t=1}^{T} \mathbb{E}_{s \sim p_0}[r(s, \pi^\star(s)) - r(s, \pi^t(s))] \leq f(T, R).$$

This assumption is usually given for free since it is implied by a rich literature in contextual bandit, including tabular Langford & Zhang (2007), linear Chu et al. (2011), general function approximation with finite action Simchi-Levi & Xu (2020), neural networks and continuous actions Krishnamurthy et al. (2019), either via optimism-based methods (UCB, Thompson sampling etc.) or non-optimism-based methods ($\epsilon$-greedy, inverse gap weighting etc.).

Now we are ready to present the algorithm and guarantee. The JSRL algorithm is summarized in Algorithm 1. For the convenience of theoretical analysis, we make some simplification by only considering curriculum case, replacing the step of EvaluatePolicy with a fixed iteration time, and set the TrainPolicy in Algorithm 1 as follows: at iteration $h$, fix the policy $\pi_{h+1:H}$ unchanged, set $\pi_h = $ ExplorationOracle_CB$(\mathcal{D})$, where the reward for contextual bandit is the cumulative reward $\sum_{t=h:H} r_t$. For concreteness, we show the pseudocode for the algorithm below.

---

**Algorithm 2** Jump-Start Reinforcement Learning for Episodic MDP with CB oracle

---

1: **Input**: guide-policy $\pi^g$, total time step $T$, horizon length $H$
2: Initialize exploration policy $\pi = \pi^g$, online dataset $\mathcal{D} = \varnothing$.
3: **for** iteration $h = H - 1, H - 2, \cdots, 0$ **do**
4:     Execute ExplorationOracle_CB for $\lceil T/H \rceil$ rounds, with the state-aciton-reward tuple for contextual bandit derived as follows: at round $t$, first gather a trajectory $\{(s_l^t, a_l^t, s_{l+1}^t, r_l^t)\}_{l \in [H-1]}$ by rolling out policy $\pi$, then take $\{s_h^t, a_h^t, \sum_{l=h}^{H} r_l^t\}$ as the state-action-reward samples for contextual bandit. Let $\pi^t$ be the executed policy at round $t$.
5:     Set policy $\pi_h = \mathsf{Unif}(\{\pi^t\}_{t=1}^T)$.
6: **end for**

---

Note that the Algorithm 2 is a special case of Algorithm 1 where the policies after current step $h$ is fixed. This coincides with the idea of Policy Search by Dynamic Programming (PSDP) in Bagnell et al. (2003). Notably, although PSDP is mainly motivated from policy learning while JSRL is motivated from efficient online exploration and fine-tuning, the following theorem follows mostly the same line as that in Bagnell (2004). For completeness we provide the performance guarantee of the algorithm as follows.

**Theorem A.3.** *Under Assumption A.1 and A.2, the JSRL in Algorithm 2 guarantees that after $T$ rounds,*

$$\mathbb{E}_{s_0 \sim p_0}[V_0^*(s_0) - V_0^\pi(s_0)] \leq C \cdot \sum_{h=0}^{H-1} f(T/H, H - h).$$

Theorem A.3 is quite general, and it depends on the choice of the exploration oracle. Below we give concrete results for tabular RL and RL with function approximation.

**Corollary A.4.** *For tabular case, when we take* ExplorationOracle_CB *as $\epsilon$-greedy, the rate achieved is $O(CH^{7/3}S^{1/3}A^{1/3}/T^{1/3})$ ; when we take* ExplorationOracle_CB *as FALCON+, the rate becomes $O(CH^{5/2}S^{1/2}A/T^{1/2})$. Here $S$ can be relaxed to the maximum state size that $\pi^g$ visits among all steps.*

The result above implies a polynomial sample complexity when combined with non-optimism exploration techniques, including $\epsilon$-greedy Langford & Zhang (2007) and FALCON+ Simchi-Levi & Xu (2020). In contrast, they both suffer from a curse of horizon without such a guide-policy.

Next, we move to RL with general function approximation.

**Corollary A.5.** *For general function approximation, when we take* ExplorationOracle_CB *as FAL-CON+, the rate becomes* $\tilde{O}(C\sum_{h=1}^{H}\sqrt{A\mathcal{E}_{\mathcal{F}}(T/H)})$ *under the following assumption.*

**Assumption A.6.** Let $\pi$ be an arbitrary policy. Given $n$ training trajectories of the form $\{(s_h^j, a_h^j, s_{h+1}^j, r_h^j)\}_{j\in[n],h\in[H]}$ drawn from following policy $\pi$ in a given MDP, according to $s_h^j \sim d_h^{\pi}, a_h^j|s_h^j \sim \pi_h(s_h), r_h^j|(s_h^j, a_h^j) \sim R_h(s_h^j, a_h^j), s_{h+1}^j|(s_h^j, a_h^j) \sim \mathbb{P}_h(\cdot|s_h^j, a_h^j)$, there exists some offline regression oracle which returns a family of predictors $\widehat{Q}_h : \mathcal{S} \times \mathcal{A} \to \mathbb{R}, h \in [H]$, such that for any $h \in [H]$, we have

$$\mathbb{E}\left[(\widehat{Q}_h(s,a) - Q_h^{\pi}(s,a))^2\right] \leq \mathcal{E}_{\mathcal{F}}(n).$$

As is shown in Simchi-Levi & Xu (2020), this assumption on offline regression oracle implies our Assumption on regret bound in Assumption A.2. When $\mathcal{E}_{\mathcal{F}}$ is a polynomial function, the above rate matches the worst-case lower bound for contextual bandit in Simchi-Levi & Xu (2020), up to a factor of $C \cdot \text{poly}(H)$.

The results above show that under Assumption A.1, one can achieve polynomial and sometimes near-optimal sample complexity up to polynomial factors of $H$ without applying Bellman update, but only with a contextual bandit oracle. In practice, we run Q-learning based exploration oracle, which may be more robust to the violation of assumptions. We leave the analysis for Q-learning based exploration oracle as a future work.

*Remark* A.7. The result generalizes to and is adaptive to the case when one has time-inhomogeneous $C$, i.e.

$$\forall h \in [H], \sup_s \frac{d_h^{\pi^\star}(\phi(s))}{d_h^{\pi^g}(\phi(s))} \leq C(h).$$

The rate becomes $\sum_{h=0}^{H-1} C(h) \cdot f(T/H, H-h)$ in this case.

In our current analysis, we heavily rely on the assumption of visitation and applied contextual bandit based exploration techniques. In our experiments, we indeed run a Q-learning based exploration algorithm which also explores the succinct states after we roll out the guide-policy. This also suggests why setting $K > 1$ and even random switching in Algorithm 1 might achieve better performance than the case of $K = 1$. We conjecture that with a Q-learning based exploration algorithm, JSRL still works even when Assumption A.1 only holds partially. We leave the related analysis for JSRL with a Q-learning based exploration oracle for future work.

### A.4.4 PROOF OF THEOREM A.3 AND COROLLARIES

*Proof.* The analysis follows a same line as Bagnell (2004). For completeness we include here. By the performance difference lemma Kakade & Langford (2002), one has

$$\mathbb{E}_{s_0 \sim d_0}[V_0^\star(s_0) - V_0^\pi(s_0)] = \sum_{h=0}^{H-1} \mathbb{E}_{s \sim d_h^\star}[Q_h^\pi(s, \pi_h^\star(s)) - Q_h^\pi(s, \pi_h(s))]. \tag{2}$$

At iteration $h$, the algorithm adopts a policy $\pi$ with $\pi_l = \pi_l^g, \forall l < h$, and fixed learned $\pi_l$ for $l > h$. The algorithm only updates $\pi_h$ during this iteration. By taking the reward as $\sum_{l=h}^{H} r_l$, this presents a contextual bandit problem with initial state distribution $d_h^{\pi^g}$, reward bounded in between $[0, H-h]$, and the expected reward for taking state action $(s, a)$ is $Q_h^\pi(s, a)$. Let $\hat{\pi}_h^\star$ be the optimal policy for this contextual bandit problem. From Assumption A.2, we know that after $T/H$ rounds at iteration $h$,

one has

$$
\sum_{h=0}^{H-1} \mathbb{E}_{s \sim d_h^\star}[Q_h^\pi(s, \pi_h^\star(s)) - Q_h^\pi(s, \pi_h(s))] \overset{(i)}{\leq} \sum_{h=0}^{H-1} \mathbb{E}_{s \sim d_h^\star}[Q_h^\pi(s, \hat{\pi}_h^\star(s)) - Q_h^\pi(s, \pi_h(s))]
$$

$$
\overset{(ii)}{=} \sum_{h=0}^{H-1} \mathbb{E}_{s \sim d_h^\star}[Q_h^\pi(\phi(s), \hat{\pi}_h^\star(\phi(s))) - Q_h^\pi(\phi(s), \pi_h(\phi(s)))]
$$

$$
\overset{(iii)}{\leq} C \cdot \sum_{h=0}^{H-1} \mathbb{E}_{s \sim d_h^{\pi g}}[Q_h^\pi(\phi(s), \hat{\pi}_h^\star(\phi(s))) - Q_h^\pi(\phi(s), \pi_h(\phi(s)))]
$$

$$
\overset{(iv)}{\leq} C \cdot \sum_{h=0}^{H-1} f(T/H, H - h).
$$

Here the inequality (i) uses the fact that $\hat{\pi}^\star$ is the optimal policy for the contextual bandit problem. The equality (ii) uses the fact that $Q, \pi$ depends on $s$ only through $\phi(s)$. The inequality (iii) comes from Assumption A.1. The inequality (iv) comes from Assumption A.2. From Equation equation 2 we know that the conclusion holds true.

When ExplorationOracle_CB is $\epsilon$-greedy, the rate in Assumption A.2 becomes $f(T, R) = R \cdot ((SA/T)^{1/3})$ Langford & Zhang (2007), which gives the rate for JSRL as $O(CH^{7/3}S^{1/3}A^{1/3}/T^{1/3})$; when we take ExplorationOracle_CB as FALCON+ in tabular case, the rate in Assumption A.2 becomes $f(T, R) = R \cdot ((SA^2/T)^{1/2})$ Simchi-Levi & Xu (2020), the final rate for JSRL becomes $O(CH^{5/2}S^{1/2}A/T^{1/2})$. When we take ExplorationOracle_CB as FALCON+ in general function approximation under Assumption A.6, the rate in Assumption A.2 becomes $f(T, R) = R \cdot (A\mathcal{E}_\mathcal{F}(T))^{1/2}$, the final rate for JSRL becomes $\tilde{O}(C \sum_{h=1}^{H} \sqrt{A\mathcal{E}_\mathcal{F}(T/H)})$. □

