# OpenReview forum: "Jump-Start Reinforcement Learning"
_ICLR.cc/2023/Conference — Submitted to ICLR 2023_

### Official Review · Reviewer_hK9Y · 2022-10-21

**Confidence:** 4
**Correctness:** 3
**Technical Novelty And Significance:** 3
**Empirical Novelty And Significance:** 3
**Recommendation:** 6

**Clarity, Quality, Novelty And Reproducibility:**

Clarity: The overall paper is clear

Quality: Good enough

Novelty: The author's choice of perspective is interesting

Reproducibility: Should be fine

**Strength And Weaknesses:**


Strength.
* The paper is clearly written and very well understood
* The method is helpful when reinforcement learning is performed for application
* Experimental results show that the method proposed by the authors has clear advantages

Weaknesses.
* The paper emphasizes the shortcomings of the value-based approach, and in fact most of the experiments are based on the AC architecture approach.
* The paper has some overclaims, mentioning "It is also compatible with any RL algorithm and can be easily combined with existing offline and/or online RL methods" several times in the paper, but it is worth noting that JSRL is only applicable to some specific scenarios, such as offline to online, and I think the authors should be more rigorous in expressing the advantages of the method.
* The results of Fig2 are intuitive, and since critic is not learned, it is natural that efficient finetuning is not possible. in conjunction with Fig7 and Fig8 in the appendix, I believe that the core problem comes from the need for reasonable policy evaluation learning at the beginning of the next phase in order to proceed well with further learning, but I find the authors' formulation in the relevant section confusing .
* The authors should have compared transfer learning and offline2online related methods instead of a few simple baseline algorithms.
* In Fig3, it looks like there is no significant advantage of gradually decreasing methods compared with the randomly chosen time step methods



**Summary Of The Paper:**

The authors propose JSRL to explore and learn by continuously using existing polices to reach a state before adopting a learnable policy. The authors emphasize that when using existing policy to select the arrival state, one can consider gradually decreasing the time step of their decisions, thus providing a more stable learning process.

**Summary Of The Review:**

I think this paper is interesting enough that I will take into account the comments of other reviewers with the results of the rebuttal to adjust my score

---

> ### Author Response · Authors · 2022-11-09
> **Initial Response to Reviewer hK9Y (1/2)**
>
> Thank you for the positive review and helpful feedback! Please let us know if the points below are sufficient clarification for your concerns.
>
> ***
>
> > The paper emphasizes the shortcomings of the value-based approach, and in fact most of the experiments are based on the AC architecture approach.
>
> The actor-critic (AC) approach is commonly the go-to value based RL approach for domains with continuous actions, and continuous action spaces are common in many practical RL tasks. Since there are infinitely many actions to choose from at each timestep, naïve approaches such as $\epsilon$-greedy with a value function do not work. In AC, the actor is responsible for sampling an action, and the critic (value function) can then evaluate the actor’s choice.
>
> AC algorithms usually combine learning a value function with learning a behavior policy. In many AC algorithms such as IQL [1] , AWAC [2], DDPG [3], etc., the actor and critic are trained in unison. This joint training regime allows for expressive algorithms that can leverage the strengths of approximate dynamic programming and supervised learning.
>
> Unfortunately, AC algorithms are susceptible to the weaknesses of both approaches. This is the motivation for Figure 2: directly fine-tuning a pre-trained policy with a randomly initialized critic is not enough to achieve higher performance. Additionally, in Figures 7 and 8, we show that even allowing the critic to “warm up” with rollouts from the pre-trained policy is still not suitable for fine-tuning.
>
> [1] Kostrikov, I., Nair, A., & Levine, S. (2021). Offline reinforcement learning with implicit q-learning. arXiv preprint arXiv:2110.06169.
>
> [2] Nair, A., Gupta, A., Dalal, M., & Levine, S. (2020). Awac: Accelerating online reinforcement learning with offline datasets. arXiv preprint arXiv:2006.09359
>
> [3] Lillicrap, T.P., Hunt, J.J., Pritzel, A., Heess, N., Erez, T., Tassa, Y., Silver, D. and Wierstra, D., 2015. Continuous control with deep reinforcement learning. arXiv preprint arXiv:1509.02971.
>
> ***
>
> > The paper has some overclaims, mentioning "It is also compatible with any RL algorithm and can be easily combined with existing offline and/or online RL methods" several times in the paper, but it is worth noting that JSRL is only applicable to some specific scenarios, such as offline to online, and I think the authors should be more rigorous in expressing the advantages of the method.
>
> Thank you for bringing this to our attention. It is true that JSRL is only applicable in scenarios where we have access to a pre-existing policy. An ideal use case of JSRL is the situation where one has access to a sub-optimal pre-existing policy, and would like to improve it with additional fine-tuning. We will rewrite our claims in the paper to accurately reflect this.
>
> ***
>
> > The results of Fig2 are intuitive, and since critic is not learned, it is natural that efficient finetuning is not possible. in conjunction with Fig7 and Fig8 in the appendix, I believe that the core problem comes from the need for reasonable policy evaluation learning at the beginning of the next phase in order to proceed well with further learning, but I find the authors' formulation in the relevant section confusing .
>
> We assume that the policy evaluation learning procedure is similar to what we call “warming up” the critic in Figures 7 and 8. This procedure involves freezing the weights of the policy, then collecting rollouts from it to train a value function. Please clarify our assumption if these are not the same. Policy evaluation learning at the beginning of the fine-tuning phase is indeed important. Our baseline IL+RL methods such as AWAC and IQL do this implicitly. The motivating example in Figure 2 is a toy task that is meant to be naïve for the purpose of illustrating why IL+RL is a difficult problem.
>
> ***
>
> > In Fig3, it looks like there is no significant advantage of gradually decreasing methods compared with the randomly chosen time step methods
>
> Thank you for pointing this out. The left half of Figure 3 depicts the results for the simpler indiscriminate grasping environment. In indiscriminate grasping, the agent is rewarded for picking up **any** object in the environment. It is true that there is no significant advantage in gradually decreasing vs. randomly chosen timesteps in this easier setting. Although, in Figure 11, we find that gradually decreasing is generally more sample efficient on the more complex instance grasping task. In instance grasping, the agent is only rewarded for picking up a **certain type** of object.

---

> > ### Author Response · Authors · 2022-11-09
> > **Initial Response to Reviewer hK9Y (2/2)**
> >
> > > The authors should have compared transfer learning and offline2online related methods instead of a few simple baseline algorithms.
> >
> > Most of our baseline algorithms are imitation and reinforcement learning methods (IL+RL), which are often also referred to as offline2online methods. **IL+RL methods usually involve pre-training on offline data, then fine-tuning the pre-trained policies online**. We do not include transfer learning methods because our goal is to use demonstrations or sub-optimal pre-existing policies to speed up RL training. Transfer learning usually implies distilling knowledge from a well performing model to another (often smaller) model, or repurposing an existing model to solve a new task. Both of these use cases are outside the scope of our work.
> >
> > Here is a list of the IL+RL baseline algorithms we used:
> >
> > * D4RL:
> >     * **AWAC** [1]: AWAC is an actor-critic method that updates the critic with dynamic programming and updates the actor such that its distribution stays close to the behavior policy that generated the offline data. Note that the AWAC paper compares against a few additional IL+RL baselines, including a few variants that use demonstrations with vanilla SAC.
> >     * **CQL** [2]: CQL is a Q-learning variant that regularizes Q-values during training to avoid the estimation errors caused by performing Bellman updates with out of distribution actions.
> >     * **IQL** [3]: IQL is an actor-critic method that completely avoids estimating the values of actions that are not seen in the offline dataset. This is a recent state-of-the-art method for the IL+RL setting we consider.
> >
> > * Simulated Robotic Grasping
> >     * **AW-Opt** [4]: AW-Opt combines insights from [1] and [5] to create a distributed actor-critic algorithm that can successfully fine-tune policies trained offline. [5] is an RL system that has been shown to scale to complex, high-dimensional robotic control from pixels, which is a much more challenging domain than common simulation benchmarks like D4RL.
> >
> > [1] Nair, A., Gupta, A., Dalal, M., & Levine, S. (2020). Awac: Accelerating online reinforcement learning with offline datasets. arXiv preprint arXiv:2006.09359
> >
> > [2] Kumar, A., Zhou, A., Tucker, G. and Levine, S., 2020. Conservative q-learning for offline reinforcement learning. Advances in Neural Information Processing Systems, 33, pp.1179-1191.
> >
> > [3] Kostrikov, I., Nair, A. and Levine, S., 2021. Offline reinforcement learning with implicit q-learning. arXiv preprint arXiv:2110.06169.
> >
> > [4] Lu, Y., Hausman, K., Chebotar, Y., Yan, M., Jang, E., Herzog, A., Xiao, T., Irpan, A., Khansari, M., Kalashnikov, D. and Levine, S., 2022, January. AW-opt: Learning robotic skills with imitation andreinforcement at scale. In Conference on Robot Learning (pp. 1078-1088). PMLR.
> >
> > [5] Kalashnikov, D., Irpan, A., Pastor, P., Ibarz, J., Herzog, A., Jang, E., Quillen, D., Holly, E., Kalakrishnan, M., Vanhoucke, V. and Levine, S., 2018, October. Scalable deep reinforcement learning for vision-based robotic manipulation. In Conference on Robot Learning (pp. 651-673). PMLR.

---

> > > ### Comment · Reviewer_hK9Y · 2022-11-18
> > > **Reply To The Authors**
> > >
> > > Dear Authors.
> > >
> > > Thank you for your detailed replies, the current ones solved my confusion. I will consider the comments of other reviewers and then further consider my decision.

---

### Official Review · Reviewer_3rP7 · 2022-10-24

**Confidence:** 4
**Correctness:** 4
**Technical Novelty And Significance:** 3
**Empirical Novelty And Significance:** 3
**Recommendation:** 8

**Clarity, Quality, Novelty And Reproducibility:**

This paper is very well written. The proof is easy to follow. The proposed method is novel.

**Strength And Weaknesses:**

The method proposed in this paper is a very general framework for efficiently and quickly learning an RL policy by utilizing the knowledge of an existing guide-policy. Any RL algorithm and policy model can be plugged into the proposed framework to improve its performance under an reasonable assumption. Strong theoretical proofs are provided to verity the advantages of this paper. From the experimental results, the improvement is significant compared to the existing methods. Different experimental setups are considered which gives a sufficient support for the theoretical result.

I understand that Assumption 4.2 is relatively weak compared to the assumptions in related works. And this assumption is also essential for the theory. But in reality, I still think it is pretty strong. It could be hard to find a guide policy which only visits all good states in feature space.

**Summary Of The Paper:**

This paper proposes a general method called Just-Start Reinforcement Learning to utilize the pre-existing policy for better learning an RL policy. Two policies, a guide-policy and a exploration-policy, are used interactively to efficiently explore the spaces. The guide-policy is used at the beginning of each training episode. And an updated exploration policy will take charge of the other steps until the end of the time horizon. This paper theoretically analyzes the upper bound on the sample complexity of JSRL is polynomial. Furthermore, this paper presents simulated experiments on a set of robotics task. The proposed method significantly outperform related IL+RL works.

**Summary Of The Review:**

This paper proposes a novel frame work for efficiently exploring the state by using existing knowledge. Sufficient theoretical and empirical analysis is provided to support the statement in this paper.

---

> ### Author Response · Authors · 2022-11-09
> **Initial Response to Reviewer 3rP7**
>
> Thank you for the positive review and helpful feedback! Please let us know if the point below is sufficient clarification for your concerns.
>
> ***
>
> > I understand that Assumption 4.2 is relatively weak compared to the assumptions in related works. And this assumption is also essential for the theory. But in reality, I still think it is pretty strong. It could be hard to find a guide policy which only visits all good states in feature space.
>
> In reality, we can relax the assumption such that it only requires that the guide policy visits states that are visited by some good policy $\pi$ (instead of the optimal policy $\pi^\star$). And we can show that JSRL can have similar performance compared to such policy $\pi$ instead of $\pi^\star$. Although the guarantee is weaker, it is more practical since we can always search from all policies that have similar visitation as the guide policy, and then identify the one with the best performance.

---

### Official Review · Reviewer_5862 · 2022-11-01

**Confidence:** 4
**Clarity, Quality, Novelty And Reproducibility:** The paper is easy to follow and writi…
**Correctness:** 3
**Technical Novelty And Significance:** 4
**Empirical Novelty And Significance:** 4
**Recommendation:** 6

**Strength And Weaknesses:**

Strengths:
- The paper is well-written.
- The problem addressed in the paper is very important problem for RL to be used widely as a learning method.
- The paper evaluates the proposed method on extensive simulated and vision-based robotic tasks.
- The upper bound on sample complexity of the proposed method is good.

Weaknesses/Questions:
- The problem of sample complexity of RL is very useful in real-world robotic tasks. I have only seen authors mentioning about real-world robotic task in the end of the paper as Future Work. Can authors discuss how the proposed method would help in real-world robotic tasks?  If authors have some small experiment on a real-world robotic task to demonstrate effectiveness of the proposed method that would make the paper more strong?
-  The guided policy induces some bias in learning the policy to learn a task. However, RL might find a better policy from scratch if the guided policy is not there? Can author comment on the possible bias induced due to the guided policy?
- The guided policy is chosen from similar domain task as the new task? How would JSRL would perform if a guided policy is from different domain than the new task? For example, can we use a guided policy of grasping on a door closing task?
- Could you please mention abbreviation of IQL? I know you guys are referring to Implicit Q-Learning. However, its good for the reader to know this terms in the paper.

**Summary Of The Paper:**

The paper addresses an interesting problem for RL i.e., sample efficiency. The author proposes a Jump-Start RL (JSRL) algorithm to leverage a prior policy of any form to give a head start for exploration in RL. The proposed algorithm rolls out a pre-existing guided policy, followed by self-improving exploration policy. The proposed method is evaluated on simulated tasks and vision-based robotic tasks. Also, the author provides the upper bound on the sample complexity for proposed method.

**Summary Of The Review:**

already mentioned above.

---

> ### Author Response · Authors · 2022-11-09
> **Initial Response to Reviewer 5862**
>
> Thank you for the helpful feedback! Please let us know if the points below are sufficient clarification for your concerns.
>
> ***
>
> >  I have only seen authors mentioning about real-world robotic task in the end of the paper as Future Work. Can authors discuss how the proposed method would help in real-world robotic tasks? If authors have some small experiment on a real-world robotic task to demonstrate effectiveness of the proposed method that would make the paper more strong?
>
> Since we did not perform real-world evaluation, we can only speculate on the benefits that JSRL would have. Important challenges for real-world robot learning are **time** and **safety**.
>
> When training from scratch, the robot’s policy will be extremely poor in early stages. This leads to many failures where the robot must be reset. Unlike simulated environments, this could be a costly procedure. With JSRL, the robot could avoid numerous early resets.
>
> We also believe that JSRL could allow for safer training rollouts since the robot would initially follow a pre-existing policy that could potentially be hard-coded to avoid safety violations.
>
>
> ***
>
> > The guided policy induces some bias in learning the policy to learn a task. However, RL might find a better policy from scratch if the guided policy is not there? Can author comment on the possible bias induced due to the guided policy?
>
> This is a great observation! It is true that the guide-policy does induce bias in the exploration-policy. This is because the initial states that the exploration-policy sees are sampled directly from the state distribution of the guide-policy.
>
> The settings where JSRL is most useful are those where RL **cannot** find a better policy from scratch. For example, consider the difficult antmaze tasks from Table 2 (medium-play, medium-diverse, large-play, large-diverse). To our knowledge, it is **not possible** for these tasks to be solved from scratch without help from demonstrations or a pre-existing policy. **We plan to demonstrate this fact by training these environments from scratch with SAC [1]**. We will update our response with the results of this experiment. Thank you for your patience!
>
> [1] Haarnoja, T., Zhou, A., Abbeel, P. and Levine, S., 2018, July. Soft actor-critic: Off-policy maximum entropy deep reinforcement learning with a stochastic actor. In International conference on machine learning (pp. 1861-1870). PMLR.
>
> ***
>
> > The guided policy is chosen from similar domain task as the new task? How would JSRL would perform if a guided policy is from different domain than the new task? For example, can we use a guided policy of grasping on a door closing task?
>
> Using guide-policies from different domains would be an interesting experiment! In Figure 14, we show how policies trained on demonstrations from easier antmaze tasks perform when used as guide-policies for harder antmaze tasks. These experiments show that using a guide-policy from a simpler task (in the same domain) with IQL+JSRL yields higher returns compared to vanilla IQL. Figure 13 depicts a similar experiment, but using our vision-based simulated robotic grasping domains.
>
> In general, we suspect that there are some **pairs** of domains where cross-domain JSRL would work well. For example, if the domains are amenable to each other, i.e., the observation and action spaces match, or there is an easy conversion between them. In Figure 13, we were able to transfer policies between antmaze environments because the action/observation spaces match, even though the mazes vary greatly. In Figure 14, we could transfer policies from indiscriminate grasping to instance grasping by modifying the pixel observations. Instance grasping uses a pixel mask for the target object, so we pass the raw pixel observation to the indiscriminate grasping policy without mask. This allows the indiscriminate grasping policy to operate in this new domain. On the other hand, using a guide-policy trained on relocate-binary-v0 for door-binary-v0 would require careful manual engineering since the observation spaces differ ([relocate](https://github.com/vikashplus/mj_envs/blob/f786982204e85b79bd921aa54ffebf3a7887de3d/mj_envs/hand_manipulation_suite/relocate_v0.py#L68) vs. [door](https://github.com/vikashplus/mj_envs/blob/f786982204e85b79bd921aa54ffebf3a7887de3d/mj_envs/hand_manipulation_suite/door_v0.py#L80)), and the tasks are much different.
>
> ***
>
> > Could you please mention abbreviation of IQL? I know you guys are referring to Implicit Q-Learning. However, its good for the reader to know this terms in the paper.
>
> Thank you for pointing this out! We will update the manuscript to include the full name and a brief description of the Implicit Q-Learning algorithm.

---

> > ### Author Response · Authors · 2022-12-13
> > **Training from Scratch with SAC**
> >
> > Thank you very much for your patience regarding our additional experiments! We aimed to demonstrate that the larger antmaze environments are too difficult to train from scratch with SAC [1]. Please see the table below (we report the results of three random seeds for each environment):
> >
> > |Environment                |SAC          |
> > |---------------------------|---------|
> > |antmaze-medium-play-v0     |0.00±0.00     |
> > |antmaze-medium-diverse-v0  |0.00±0.00     |
> > |antmaze-large-play-v0      |0.00±0.00     |
> > |antmaze-large-diverse-v0   |0.00±0.00     |
> >
> >
> >
> > [1] Haarnoja, T., Zhou, A., Abbeel, P. and Levine, S., 2018, July. Soft actor-critic: Off-policy maximum entropy deep reinforcement learning with a stochastic actor. In International conference on machine learning (pp. 1861-1870). PMLR.

---

### Official Review · Reviewer_WwEP · 2022-11-02

**Confidence:** 3
**Correctness:** 2
**Technical Novelty And Significance:** 2
**Empirical Novelty And Significance:** 2
**Recommendation:** 3

**Clarity, Quality, Novelty And Reproducibility:**

The paper is relatively easy to read. However, in the theoretical analysis, some languages appear imprecise.

1. For Theorem 4.3, the authors wrote “To achieve a polynomial bound for JSRL, it suffices to take TrainPolicy as $\epsilon$-greedy.”

The TrainPolicy procedure updates a policy and a Q function. $\epsilon$-greedy normally refers to an exploration method for policy. The authors should provide a more precise explanation of what an $\epsilon$-greedy policy update procedure is.

2. The distribution mismatch coefficient in Assumption 4.2 uses undefined quantities $d$.

3. In Section 4.2, I think it should be $H_i \in \{1, 2, \cdots, H\}$.


**Strength And Weaknesses:**

Strength: JSRL is well-motivated to address the sample complexity requirement in training RL agents from scratch. The proposed algorithm is easy to understand and empirical evaluations show some effectiveness, especially in the low demo data regime.

Weakness:
The empirical evaluation section could be improved. In Table 2, three baseline methods were not tested on low demo data regimes, only the 1 million standard setting. As some of them achieve better results than JSRL in the standard setting, we need to see the comparisons in the low data setting to evaluate how effective JSRL really is.

Moreover, the improvement margins JSRL has over baseline methods are relatively small. In fact, in Table 1, for Instance Grasping, the result confidence intervals of 20 and 20k demos overlap with those of AW-Opt. The improvement in D4RL tasks is also small. So I am not convinced JSRL, in its current form, is really better than existing baselines.

There are some imprecise writings in the theoretical analysis, see my questions in the next section.


**Summary Of The Paper:**

In this paper, the authors propose Jump-Start Reinforcement Learning (JSRL) which utilizes a pre-trained guide policy to form a curriculum of starting states for a different exploration policy. Theoretical analysis shows that with a properly chosen training and evaluation algorithm, JSRL achieves a polynomial sample complexity. Empirical evaluations on D4RL and vision-based robotic tasks are provided to show the effectiveness of JSRL.

**Summary Of The Review:**

The major weakness is the evaluation results, which did not convince me that JSRL is stronger than baseline IL + RL methods.

---

> ### Author Response · Authors · 2022-11-09
> **Initial Response to Reviewer WwEP**
>
> Thank you for the helpful feedback! Please let us know if the points below are sufficient clarification for your concerns.
>
> ***
>
> > The empirical evaluation section could be improved. In Table 2, three baseline methods were not tested on low demo data regimes, only the 1 million standard setting. As some of them achieve better results than JSRL in the standard setting, we need to see the comparison in the low data setting to evaluate how effective JSRL really is
>
> We encountered difficulty reproducing the baselines for AWAC, BC, and CQL, so we used the results reported in [1] for the full demonstration dataset setting (1 million transitions). The remaining rows of the table are left blank since [1] did not ablate the size of the demonstration dataset. We then used IQL as the base RL algorithm for JSRL since it was the strongest imitation and reinforcement learning (IL+RL) baseline.
>
> We are working on running these baselines on the low demo data regime and we plan to update our response and the paper draft when the results are ready. Thank you for your patience!
>
> [1] Kostrikov, I., Nair, A. and Levine, S., 2021. Offline reinforcement learning with implicit q-learning. arXiv preprint arXiv:2110.06169.
>
> ***
>
> > Moreover, the improvement margins JSRL has over baseline methods are relatively small. In fact, in Table 1 for Instance Grasping, the result confidence intervals of 20 and 20k demos overlap with those of AW-Opt.
>
> While the final performances of AW-Opt and JSRL are similar for the instance grasping task, Figure 11 demonstrates that JSRL is more sample efficient. For example, in the 20 demonstration regime, JSRL converges to a higher score in ~30K fewer training steps. Having a small demonstration dataset size is representative of real world robot learning where acquiring successful demonstrations is expensive. Therefore, JSRL’s ability to achieve high performance quickly, with a small amount of demonstrations, is particularly important.
>
> ***
>
> > The improvement in D4RL tasks is also small
>
> Given access to the full demonstration dataset (one million transitions), the improvement between IQL and IQL+JSRL is indeed small. Although, as we reduce the number of demonstration transitions, we see that IQL+JSRL can achieve far higher performance with access to less data.
>
> For example, consider the two most difficult antmaze environments: antmaze-large-play-v0 and antmaze-large-diverse-v0. The only setting where IQL can achieve reasonable performance on either environment is when it has access to the full demonstration dataset of one million transitions. On the other hand, IQL+JSRL can achieve medium performance with access to 10x fewer demonstration transitions (Table 2).
>
>
> ***
>
> > The TrainPolicy procedure updates a policy and a Q function. $\epsilon$-greedy normally refers to an exploration method for policy. The authors should provide a more precise explanation of what an $\epsilon$-greedy policy update procedure is.
>
> We apologize for the confusing notation. TrainPolicy refers to the combined process of selecting an action with the exploration-policy and updating the policy/value function using a standard Bellman update. We chose to formulate it in this way so that the general algorithm would be compatible with the theoretical analysis in section 4.3. We will add a more precise explanation in the appendix and update the paper accordingly.
>
> ***
>
> > The distribution mismatch coefficient in Assumption 4.2 uses undefined quantities $d$
>
> The variable $d$ is the marginalized state occupancy distribution in step $h$ when we follow policy $\pi$, which is only defined in the Appendix in the current draft. We will include its formal definition above Assumption 4.2 in our revised manuscript.
>
> ***
>
> > In section 4.2, I think it should be $H_i \in 1,2, \cdots, H$
>
> Thank you for spotting this mistake! We will fix it in our revised manuscript.

---

> > ### Author Response · Authors · 2022-12-13
> > **Low Demo Data Baselines**
> >
> > Thank you very much for your patience regarding our additional experiments! Please see the table below for results on the baseline IL+RL algorithms in the low demo data regime. For AWAC and CQL, we report the mean and standard deviation over three random seeds. For BC, we report the performance of a single random seed. The results for IQL+JSRL are the "IQL+JSRL (Curriculum)" results from our manuscript in Table 2, which report the mean and standard deviation over twenty random seeds. Overall, we see that IQL+JSRL outperforms the baseline IL+RL algorithms in the low demo data regime.
> >
> >
> > ***
> >
> >
> > |Environment                |Num Demos|AWAC          |CQL           |BC       |IQL+JSRL
> > |---------------------------|---------|--------------|--------------|---------|--------------|
> > |antmaze-umaze-v0           |1000     |0.00±0.00     |0.00±0.00     |0.00     |**15.6±19.9**
> > |                           |10000    |0.00±0.00     |0.00±0.00     |1.00    |**71.7±14.5**
> > |                           |100000   |0.00±0.00     |0.00±0.00     |62.00    |**93.7±4.2**
> > |                           |1000000  |**93.67±1.89**    |**64.33±45.58**   |61.00    |**98.1±1.4**
> > |antmaze-umaze-diverse-v0   |1000     |0.00±0.00     |0.00±0.00     |0.00     |**3.1±8.0**
> > |                           |10000    |0.00±0.00     |0.00±0.00     |1.00     |**72.6±12.2**
> > |                           |100000   |0.00±0.00     |0.00±0.00     |13.00    |**81.3±23.0**
> > |                           |1000000  |46.67±3.68    |0.50±0.50     |**80.00**    |**88.6±16.3**
> > |antmaze-medium-play-v0     |1000     |0.00±0.00     |0.00±0.00     |0.00     |0.00±0.00
> > |                           |10000    |0.00±0.00     |0.00±0.00     |0.00     |**16.7±12.9**
> > |                           |100000   |0.00±0.00     |0.00±0.00     |0.00     |**86.7±3.7**
> > |                           |1000000  |0.00±0.00     |0.00±0.00     |0.00     |**91.1±3.9**
> > |antmaze-medium-diverse-v0  |1000     |0.00±0.00     |0.00±0.00     |0.00     |0.00
> > |                           |10000    |0.00±0.00     |0.00±0.00     |0.00     |**16.6±11.7**
> > |                           |100000   |0.00±0.00     |0.00±0.00     |0.00     |**81.5±18.8**
> > |                           |1000000  |0.00±0.00     |0.00±0.00     |0.00     |**93.1±3.1**
> > |antmaze-large-play-v0      |1000     |0.00±0.00     |0.00±0.00     |0.00     |0.00
> > |                           |10000    |0.00±0.00     |0.00±0.00     |0.00     |**0.1±0.2**
> > |                           |100000   |0.00±0.00     |0.00±0.00     |0.00     |**36.3±16.4**
> > |                           |1000000  |0.00±0.00     |0.00±0.00     |0.00     |**62.9±11.3**
> > |antmaze-large-diverse-v0   |1000     |0.00±0.00     |0.00±0.00     |0.00     |0.00
> > |                           |10000    |0.00±0.00     |0.00±0.00     |0.00     |**0.1±0.2**
> > |                           |100000   |0.00±0.00     |0.00±0.00     |0.00     |**34.4±23.0**
> > |                           |1000000  |0.00±0.00     |0.00±0.00     |0.00     |**68.3±8.8**
> > |door-binary-v0             |100      |**0.07±0.11**     |0.00±0.00     |0.00     |**0.4±1.8**
> > |                           |1000     |**0.41±0.58**     |0.00±0.00     |0.00     |**0.7±1.0**
> > |                           |10000    |1.93±2.72     |**12.24±24.47**   |0.00     |4.3±8.4
> > |                           |100000   |**17.26±20.09**   |**8.28±19.94**    |0.00     |**28.5±19.5**
> > |pen-binary-v0              |100      |3.13±4.43     |**31.46±9.99**    |0.00     |**24.3±12.1**
> > |                           |1000     |1.43±1.10     |**54.50±0.00**    |0.00     |36.7±7.9
> > |                           |10000    |2.21±1.30     |**51.36±4.34**    |0.00     |**44.3±6.2**
> > |                           |100000   |1.23±1.08     |**59.58±1.43**    |0.00     |**62.6±3.6**
> > |relocate-binary-v0         |100      |0.00±0.00     |0.00±0.00     |0.00     |**0.0±0.1**
> > |                           |1000     |**0.01±0.01**     |0.00±0.00     |0.00     |**0.0±0.1**
> > |                           |10000    |0.00±0.00     |**1.18±2.70**     |0.00     |**0.6±1.6**
> > |                           |100000   |0.00±0.00     |**4.44±6.36**     |0.00     |0.0±0.1

---

### Author Response · Authors · 2022-11-19
**Updated Manuscript**

We have updated our submission to implement the insights suggested by Reviewers WwEP, 5682, and hK9Y.

At this time, we are still working on running the additional experiments. Thank you all for your patience!

---

### Decision · Program_Chairs · 2023-01-20

**Decision:**

Reject

**Justification For Why Not Higher Score:**

The general idea of somehow getting the agent close to the goal in the beginning of the learning process and to rely less and less on that mechanism is well known. The idea here is a practically feasible way, that has the additional benefit of allowing to use the data generated by getting to the starting point as well. However, the current results are insufficient to back this claim, the promised additional results were not delivered.

*Note:* The additional results came too late (after the rebuttal phase, after the AC-reviewer meeting, and even after the AC report deadline). To keep things fair towards other authors we cannot consider the additional results.

**Justification For Why Not Lower Score:**

N/A

**Metareview: Summary, Strengths And Weaknesses:**

Summary:
The paper proposes a new form of war-starting RL: a guide-policy that can roughly solve the task is used for the initial part of an episode bringing the agent to a point from whereon it can solve the task itself. The influence of the guide-policy can then be decayed. The approach is bench-marked experimentally.

Strengths:
- The paper is well written and very clear
- The paper tackles an important problem
- The idea is practically applicable

Weaknesses:
- The general idea is well known
- Concerns of biasing the solution
- The results are somewhat inconclusive
- The promised results in the low data setting and SAC were not delivered

Small additional comments from re-reading the paper:
- H_i \in {..., H} is ugly
- Sect. 4.2 mentions the "combined policy \pi", which isn't mentioned in in the algorithm. Doesn't sound like a combined policy but rather a switching one.

*Note:* the additional results came in after the AC report deadline


**Summary Of Ac-Reviewer Meeting:**

Scheduling a meeting with everybody present was impossible. 1 reviewer never reacted, 1 was not available when the majority was, 1 said s/he would be available, in the end only reviewer WwEP attended.
This reviewer's main concern is still not addressed. One of the main claims, i.e., improvement in the low data volume setting  is not substantiated. In the large data volume case the difference is not significant. The authors proposed additional results but never delivered them.